# Traversing Between Modes in Function Space for Fast Ensembling

## Abstract

Deep ensemble is a simple yet powerful way to improve the performance of deep neural networks. Under this motivation, recent works on mode connectivity have shown that parameters of ensembles are connected by low-loss subspaces, and one can efficiently collect ensemble parameters in those subspaces. While this provides a way to efficiently train ensembles, for inference, one should still execute multiple forward passes using all the ensemble parameters, which often becomes a serious bottleneck for real-world deployment. In this work, we propose a novel framework to reduce such costs. Given a low-loss subspace connecting two modes of a neural network, we build an additional neural network predicting outputs of the original neural network evaluated at a certain point in the low-loss subspace. The additional neural network, what we call a "bridge", is a lightweight network taking minimal features from the original network, and predicting outputs for the low-loss subspace without forward passes through the original network. We empirically demonstrate that we can indeed train such bridge networks and significantly reduce inference costs with the help of the bridge networks.

## 1 Introduction

Deep Ensemble (DE) (Lakshminarayanan et al., 2017) is a simple algorithm to improve both predictive accuracy and uncertainty calibration of deep neural networks, where a neural network is trained multiple times using the same data but with different random seeds. Due to this randomness, the parameters obtained from the multiple training runs reach different local optima, called modes, on the loss surface (Fort et al., 2019). These parameters represent a set of diverse functions serving as an effective approximation for Bayesian Model Averaging (BMA) (Wilson and Izmailov, 2020).

An apparent drawback of DE is that it requires multiple training runs. This cost can be huge especially for large-scale settings for which parallel training is not feasible. Garipov et al. (2018); Draxler et al. (2018) showed that modes in the loss surface of a deep neural network are connected by relatively simple low-dimensional subspaces where every parameter on those subspaces retains low training error, and the parameters along those subspaces are good candidates for ensembling. Based on this observation, Garipov et al. (2018); Huang et al. (2017) proposed algorithms to quickly construct deep ensembles without having to run multiple independent training runs.

While the fast ensembling methods based on mode connectivity reduce training costs, they do not address another important drawback of DE; *the inference cost*. One should still execute multiple forward passes using all the parameters collected for ensemble, and this cost often becomes critical for a real-world scenario, where the training is done in a resource-abundant setting with plenty of computation time, but for the deployment, the inference should be done in a resource-limited environment. For such settings, reducing the inference cost is much more important than reducing the training cost.

In this paper, we propose a novel approach to scale up DE by reducing inference cost. We start from an assumption; if two modes in an ensemble are connected by a simple subspace, we can predict the outputs corresponding to the parameters on the subspace using *only the outputs computed from the modes*. In other words, we can predict the outputs evaluated at the subspace without having to forward the actual parameters on the subspace through the network. If this is indeed possible, for instance, given two modes, we can approximate an ensemble of three models consisting of

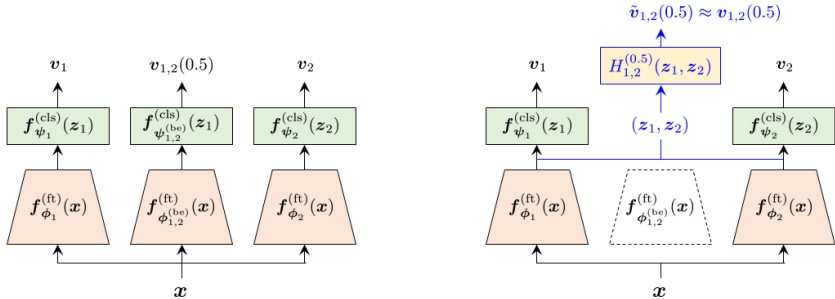

**Figure 1:** Comparing ensembles with a Bezier curve (left) and a type II bridge network (right).

parameters collected from three different locations (one from a subspace connecting two modes, and two from each mode) with only two forward passes and a small auxiliary forward pass.

We show that we can actually implement this idea using an additional lightweight network whose inference cost is relatively low compared to that of the original neural network. This additional network, what we call a "bridge network", takes some features from the original neural network, (e.g., features from the penultimate layer), and directly predict the outputs computed from the connecting subspace. In other words, the bridge network lets us travel between modes in the *function space*.

We present two types of bridge networks depending on the number of modes involved in prediction, network architectures for bridge networks, and training procedures. Through empirical validation on various image classification benchmarks, we show that 1) bridge networks can predict outputs of connecting subspaces quite accurately with minimal computation cost, and 2) DEs augmented with bridge networks can significantly reduce inference costs without big sacrifice in performance.

## 2 PRELIMINARIES

### 2.1 PROBLEM SETUP

In this paper, we discuss the $K$-way classification problem taking $D$-dimensional inputs. A classifier is constructed with a deep neural network $f_{\boldsymbol{\theta}} : \mathbb{R}^D \to \mathbb{R}^K$ which is decomposed into a *feature extractor* $f_{\boldsymbol{\phi}}^{(\text{ft})} : \mathbb{R}^D \to \mathbb{R}^{D_{\text{ft}}}$ and a *classifier* $f_{\boldsymbol{\psi}}^{(\text{cls})} : \mathbb{R}^{D_{\text{ft}}} \to \mathbb{R}^K$, i.e., $f_{\boldsymbol{\theta}}(\boldsymbol{x}) = f_{\boldsymbol{\psi}}^{(\text{cls})} \circ f_{\boldsymbol{\phi}}^{(\text{ft})}(\boldsymbol{x})$. Here, $\boldsymbol{\phi} \in \boldsymbol{\Phi}$ and $\boldsymbol{\psi} \in \boldsymbol{\Psi}$ denote the parameters for the feature extractor and classifier, respectively, $\boldsymbol{\theta} = (\boldsymbol{\phi}, \boldsymbol{\psi}) \in \boldsymbol{\Theta}$, and $D_{\text{ft}}$ is the dimension of the feature. An output from the classifier corresponds to a class probability vector.

### 2.2 FINDING LOW-LOSS SUBSPACES

While there are few low-loss subspaces that are known to connect modes of deep neural networks, in this paper, we focus on *Bezier curves* as suggested in (Garipov et al., 2018). Let $\boldsymbol{\theta}_i$ and $\boldsymbol{\theta}_j$ be two parameters (usually corresponding to modes) of a neural network. The quadratic Bezier curve between them is defined as

$$\left\{ (1-r)^2 \boldsymbol{\theta}_i + 2r(1-r)\boldsymbol{\theta}_{i,j}^{(\text{be})} + r^2 \boldsymbol{\theta}_j \mid r \in [0,1] \right\}, \tag{1}$$

where $\boldsymbol{\theta}_{i,j}^{(\text{be})}$ is a *pin-point* parameter characterizing the curve. Based on this curve paramerization, a low-loss subspace connecting $(\boldsymbol{\theta}_i, \boldsymbol{\theta}_j)$ is found by minimizing the following loss w.r.t. $\boldsymbol{\theta}_{i,j}^{(\text{be})}$,

$$\int_0^1 \mathcal{L}\left( \boldsymbol{\theta}_{i,j}^{(\text{be})}(r) \right) \mathrm{d}r, \tag{2}$$

where $\boldsymbol{\theta}_{i,j}^{(\text{be})}(r)$ denotes the point at the position $r$ of the curve,

$$\boldsymbol{\theta}_{i,j}^{(\text{be})}(r) = (1-r)^2 \boldsymbol{\theta}_i + 2r(1-r)\boldsymbol{\theta}_{i,j}^{(\text{be})} + r^2 \boldsymbol{\theta}_j, \tag{3}$$

and $\mathcal{L} : \boldsymbol{\Theta} \to \mathbb{R}$ is the loss function evaluating parameters (e.g., cross entropy). Since the integration above is usually intractable, we instead minimize the stochastic approximation:

$$\mathbb{E}_{r \sim \mathcal{U}(0,1)} \left[ \mathcal{L} \left( \boldsymbol{\theta}_{i,j}^{(\mathrm{be})}(r) \right) \right], \tag{4}$$

where $\mathcal{U}(0, 1)$ is the uniform distribution on $[0, 1]$. For more detailed procedure for the Bezier curve training, please refer to Garipov et al. (2018).

## 2.3 ENSEMBLES WITH BEZIER CURVES

Let $\{\boldsymbol{\theta}_1, \ldots, \boldsymbol{\theta}_m\}$ be a set of parameters independently trained as a deep ensemble. Then, for each pair $(\boldsymbol{\theta}_i, \boldsymbol{\theta}_j)$, we can construct a low-loss Bezier curve. Since all the parameters along those Bezier curves achieve low loss, we can actually add them to the ensemble for improved performance. For instance, choosing $r = 0.5$, we can collect $\boldsymbol{\theta}_{i,j}^{(\mathrm{be})}(0.5)$ for all $(i, j)$ pairs, and construct an ensembled predictor as

$$\frac{1}{m + \binom{m}{2}} \left( \sum_{i=1}^{m} f_{\boldsymbol{\theta}_i}(\boldsymbol{x}) + \sum_{i<j} f_{\boldsymbol{\theta}_{i,j}^{(\mathrm{be})}(0.5)}(\boldsymbol{x}) \right). \tag{5}$$

While this strategy provide an effective way to increase the number of ensemble members, for inference, an additional $\mathcal{O}(m^2)$ number of forward passes are required. Our primary goal in this paper is to reduce this additional cost by bypassing the direct forward passes with $\boldsymbol{\theta}_{i,j}^{(\mathrm{be})}(r)$.

# 3 MAIN CONTRIBUTION

In this section, we present a novel method that directly predicts outputs of neural networks evaluated at parameters on Bezier curves without actual forward passes with them.

## 3.1 BRIDGE NETWORKS

Let us first recall our key assumption stated in the introduction; if two modes in an ensemble are connected by a simple low-loss subspace (Bezier curve), then we can predict the outputs corresponding to the parameters on the subspace using only the information obtained from the modes. The intuition behind this assumption is that, since the parameters are connected with a simple curve, the corresponding outputs may also be connected via a relatively simple mapping which is far less complex than the original neural network. If such mapping exists, we may learn them via a lightweight neural network.

More specifically, let $\boldsymbol{z}_i := f_{\boldsymbol{\phi}_i}^{(\mathrm{ft})}(\boldsymbol{x})$ and $\boldsymbol{v}_i := f_{\boldsymbol{\theta}_i}(\boldsymbol{x}) = f_{\boldsymbol{\psi}_i}^{(\mathrm{cls})}(\boldsymbol{z}_i)$ for $i \in \{1, \ldots, m\}$. Let $\boldsymbol{v}_{i,j}(r) := f_{\boldsymbol{\theta}_{i,j}^{(\mathrm{be})}(r)}(\boldsymbol{x})$. In order to use $\boldsymbol{v}_{i,j}(r)$ with $\boldsymbol{v}_i$ to get an ensemble, we should forward $\boldsymbol{x}$ through $f_{\boldsymbol{\theta}_{i,j}^{(\mathrm{be})}(r)}$, starting from the bottom layer. Instead, we *reuse* $\boldsymbol{z}_i$ to predict $\boldsymbol{v}_{i,j}(r)$ with a lightweight neural network. We call such lightweight neural network a "bridge network", since it lets us directly move from $\boldsymbol{v}_i$ to $\boldsymbol{v}_{i,j}(r)$ in the function space, not through the actual parameter space. A bridge network is usually constructed with a Convolutional Neural Network (CNN) whose inference cost is much lower than that of $f_{\boldsymbol{\theta}_i}$.

From the following, we introduce two types of bridge networks depending on the number of modes involved in the computation.

**Type I bridge networks** A type I bridge network $h_{i,j}^{(r)}$ takes a feature $\boldsymbol{z}_i$ from only one mode, and predicts $\boldsymbol{v}_{i,j}(r)$ as

$$\boldsymbol{v}_{i,j}(r) \approx \tilde{\boldsymbol{v}}_{i,j}(r) = h_{i,j}^{(r)}(\boldsymbol{z}_i). \tag{6}$$

A type I bridge network can be constructed between any pair of connected modes $(\boldsymbol{\theta}_i, \boldsymbol{\theta}_j)$ and an ensembled prediction for specific mode $\boldsymbol{\theta}_i$ with its Bezier parameter $\boldsymbol{\theta}_{i,j}^{(\mathrm{be})}$ can be approximated as

$$\frac{1}{2} \left( \boldsymbol{v}_i + h_{i,j}^{(r)}(\boldsymbol{z}_i) \right), \tag{7}$$

whose inference cost is nearly identical to that of $\boldsymbol{v}_i$ (nearly single forward pass). One can also connect $\boldsymbol{\theta}_i$ with multiple modes $\{\boldsymbol{\theta}_{j_1}, \ldots, \boldsymbol{\theta}_{j_k}\}$, learn bridge networks between $(i, j_1), \ldots, (i, j_k)$, and construct an ensemble

$$\frac{1}{1+k}\left(\boldsymbol{v}_i + \sum_{j=1}^{k} h_{i,j_k}^{(r)}(\boldsymbol{z}_i)\right). \quad (8)$$

Still, since the costs for $h_{i,j_k}^{(r)}$s are far lower than $\boldsymbol{v}_i$, the inference cost does not significantly increase.

**Type II bridge networks**  A type II bridge network between $(\boldsymbol{\theta}_i, \boldsymbol{\theta}_j)$ takes two features $(\boldsymbol{z}_i, \boldsymbol{z}_j)$ to predict $\boldsymbol{v}_{i,j}(r)$.

$$\boldsymbol{v}_{i,j}(r) \approx \tilde{\boldsymbol{v}}_{i,j}(r) = H_{i,j}^{(r)}(\boldsymbol{z}_i, \boldsymbol{z}_j). \quad (9)$$

An ensembled prediction with the type II bridge network is then constructed as

$$\frac{1}{3}\left(\boldsymbol{v}_i + \boldsymbol{v}_j + H_{i,j}^{(r)}(\boldsymbol{z}_i, \boldsymbol{z}_j)\right), \quad (10)$$

where we construct an ensemble of three models with effectively two forward passes (for $\boldsymbol{v}_i$ and $\boldsymbol{v}_j$). Similar to the type I bridge networks, we may construct multiple bridges between a single curves and use them together for an ensemble. Fig. 1 presents a schematic diagram comparing forward passes of ensembles with-/without a type II bridge network.

---

**Algorithm 1** Training bridge networks

---

**Require:** Training dataset $\mathcal{D}$, a pair of parameters $(\boldsymbol{\theta}_1, \boldsymbol{\theta}_2)$ and corresponding Bezier parameter $\boldsymbol{\theta}_{1,2}^{(\mathrm{be})}$, a bridge network $h_{1,2}^{(r)}$ (type I) or $H_{1,2}^{(r)}$ (type II) with parameters $\boldsymbol{\omega}$, learning rate $\eta$, a regularization scale $\lambda$, a mixup coefficient $\alpha$.

Initialize $\boldsymbol{\omega}$.
**while** not converged **do**
    Sample a mini-batch $\mathcal{B} \sim \mathcal{D}$.
    **for** $i = 1, \ldots, |\mathcal{B}|$ **do**
        Take the input $\boldsymbol{x}_i$ from $\mathcal{B}$.
        $\boldsymbol{x}_i \leftarrow \mathrm{mixup}(\boldsymbol{x}_i, \alpha)$
        $\boldsymbol{z}_1 \leftarrow f_{\boldsymbol{\phi}_1}^{(\mathrm{ft})}(\boldsymbol{x}_i), \boldsymbol{v}_1 \leftarrow f_{\boldsymbol{\psi}_1}^{(\mathrm{cls})}(\boldsymbol{z}_1)$.
        $\boldsymbol{v}_{1,2}(r) \leftarrow f_{\boldsymbol{\theta}_{1,2}^{(\mathrm{be})}(0.5)}(\boldsymbol{x}_i)$.
        **if** type I **then**
            $\tilde{\boldsymbol{v}}_{1,2}(r) \leftarrow h_{1,2}^{(0.5)}(\boldsymbol{z}_1; \boldsymbol{\omega})$.
        **else**
            $\boldsymbol{z}_2 \leftarrow f_{\boldsymbol{\phi}_2}^{(\mathrm{ft})}(\boldsymbol{x}_i), \boldsymbol{v}_2 \leftarrow f_{\boldsymbol{\psi}_2}^{(\mathrm{cls})}(\boldsymbol{z}_2)$.
            $\tilde{\boldsymbol{v}}_{1,2}(r) = H_{1,2}^{(0.5)}(\boldsymbol{z}_1, \boldsymbol{z}_2; \boldsymbol{\omega})$.
        **end if**
        $\ell_i \leftarrow D_{\mathrm{KL}}(\boldsymbol{v}_{1,2}(0.5) \| \tilde{\boldsymbol{v}}_{1,2}(0.5))$
            $- \lambda D_{\mathrm{KL}}(\boldsymbol{v}_1 \| \tilde{\boldsymbol{v}}_{1,2}(0.5))$.
    **end for**
    $\boldsymbol{\omega} \leftarrow \boldsymbol{\omega} - \eta \nabla_{\boldsymbol{\omega}} \frac{1}{|\mathcal{B}|} \sum_i \ell_i$.
**end while**
**return** $\boldsymbol{\omega}$.

---

### 3.2 LEARNING BRIDGE NETWORKS

**Fixing a position $r$ on Bezier curves**  In the definition of the bridge networks above, we fixed the value $r$. In principle, we may parameterize the bridge networks to take $r$ as an additional input to predict $\boldsymbol{v}_{i,j}(r)$ for any $r \in [0, 1]$, but we found this to be ineffective due to the difficulty of learning all the outputs corresponding to arbitrary $r$ values. Moreover, as we empirically observed in Fig. 2, the ensembling with Bezier parameters are most effective with $r = 0.5$, and adding additional parameters evaluated at different $r$ values does not significantly improve the performance. To this end, we fix $r = 0.5$ and aim to learn bridge networks predicting $\boldsymbol{v}_{i,j}(0.5)$ throughout the paper.

**Training procedure**  Let $\{\boldsymbol{\theta}_1, \ldots, \boldsymbol{\theta}_m\}$ be a set of parameters in an ensemble. Given a set of Bezier parameters $\{\boldsymbol{\theta}_{i,j}^{(\mathrm{be})}\}$ connecting them, we learn bridge networks (either type I or II) for each Bezier curve. The training procedure is straightforward. We first minimize the Kullback-Leibler divergence between the actual output from the Bezier parameters and the prediction made from the bridge network. It makes the bridge network imitate the original function defined by the Bezier parameters in the same manner as a conventional knowledge distillation (Hinton et al., 2015). In addition, we also maximize the Kullback-Leibler divergence between the base prediction and the bridge prediction to regularize the bridge to predict differently from the base model. Such regularization is quite important, when the training error of the base model is near zero; the base network and the target network (the one on the Bezier curve) will produce almost identical outputs. Further, we apply the mixup (Zhang et al., 2018) method to explore more diverse responses, preventing the bridge from learning to just copy the outputs of the base model. Refer to Algorithm 1 for the detailed training procedure.

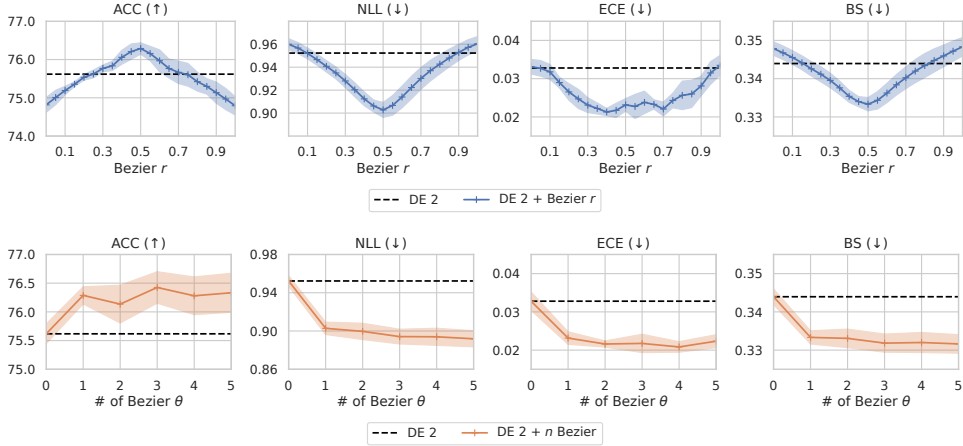

**Figure 2:** Performance of an ensemble of two modes and a parameter from the Bezier curve connecting them, evaluated for ResNet-32×4 on CIFAR-100. Here, $r \in (0, 1)$ denotes a position on the curve. **Top** row shows ensemble performances when one member from Bezier curve $r$ is added to DE-2. **Bottom** row shows ensemble performances when members are sequentially added to DE-2 from Bezier curve. For accuracy, higher is the better, and for NLL, ECE and BS, lower is the better.

## 4 RELATED WORKS

**Mode connectivity**  The geometric properties of deep neural networks' loss surfaces have been studied, and one notable property is the mode connectivity (Garipov et al., 2018; Draxler et al., 2018); there exists a simple path between modes of a neural network on which the network retains low training error along that path. From this, fast ensembling methods that collect ensemble members on the mode-connecting-paths have been proposed (Huang et al., 2017; Garipov et al., 2018). Extending this idea, Izmailov et al. (2020) approximated the posteriors of Bayesian neural nets via the low-loss subspace and used them for BMA. Wortsman et al. (2021) also presented a method for further improving performance by ensembling over the subspaces.

**Efficient ensembling**  Despite the superior performance of DE (Lakshminarayanan et al., 2017; Ovadia et al., 2019), it suffers from additional computation costs for both the training and the inference. There have been several works that reduce the computational burden in training by collecting ensemble members efficiently (Huang et al., 2017; Garipov et al., 2018; Benton et al., 2021), but they did not consider inference costs that arose from multiple forward passes. On the other hand, there also exist inference-efficient ensembling methods by sharing parameters (Wen et al., 2020; Dusenberry et al., 2021) or sharing representations (Lee et al., 2015; Siqueira et al., 2018; Antoran et al., 2020; Havasi et al., 2021). In particular, Antoran et al. (2020) and Havasi et al. (2021) presented the methods to obtain an ensemble prediction by a single forward pass. Nevertheless, these methods do not scale well for complex large-scale datasets or require large network capacity.

## 5 EXPERIMENTS

In this section, we are going to answer the following three big questions:

- Do bridge networks really learn to predict the outputs of a function from the Bezier curves?
- How much ensemble gain we obtain via bridge networks with lower computational complexity?
- How many bridge networks do we have to make in order to achieve certain ensemble performance?

We sequentially answer them in Sections 5.2 to 5.4 with empirical validation.

### 5.1 SETUP

**Datasets and networks**  We evaluate the proposed bridge networks on various image classification benchmarks, including CIFAR-10, CIFAR-100, Tiny ImageNet, and ImageNet datasets. Throughout the experiments, we use the family of residual networks introduced in He et al. (2016) as a base

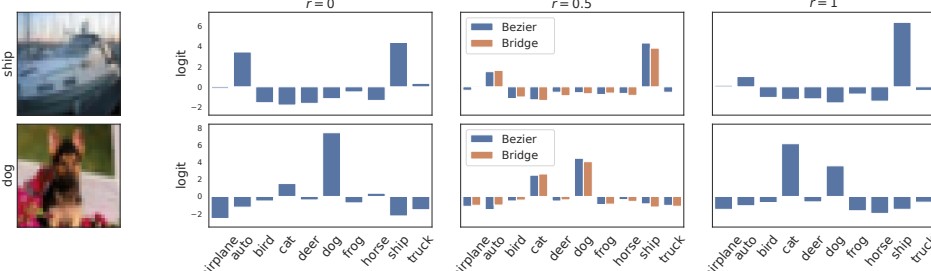

**Figure 3:** Bar plots in the third column depict whether the bridge network (orange) outputs the same logit values as the base model with the Bezier parameters (blue) for given test inputs displayed in the first column. We also depict the predicted logits from the base model with $\theta_1$ and $\theta_2$ in the second and fourth columns, respectively. Additional results are available in Fig. 6.

**Table 1:** $R^2$ scores quantify how similar the following models to the target function defined with Bezier parameters $\theta_{1,2}^{(be)}(0.5)$ are in output probabilities; 'Type I/II Bridge', 'Other Type I/II Bridge', and 'Other Bezier'. Refer to the main text in Section 5.2 for a detailed description for each model. All values are measured on the test split of each dataset.



**(a)** CIFAR-10

| Model | $R^2$ ($\uparrow$) |
|---|---|
| Type I Bridge | **0.910** $\pm$0.003 |
| Other Type I Bridge | 0.885 $\pm$0.003 |
| Type II Bridge | **0.924** $\pm$0.002 |
| Other Type II Bridge | 0.895 $\pm$0.002 |
| Other Bezier | 0.871 $\pm$0.005 |

**(b)** CIFAR-100

| Model | $R^2$ ($\uparrow$) |
|---|---|
| Type I Bridge | **0.784** $\pm$0.005 |
| Other Type I Bridge | 0.741 $\pm$0.005 |
| Type II Bridge | **0.814** $\pm$0.002 |
| Other Type II Bridge | 0.752 $\pm$0.003 |
| Other Bezier | 0.726 $\pm$0.003 |

**(c)** Tiny ImageNet

| Model | $R^2$ ($\uparrow$) |
|---|---|
| Type I Bridge | **0.746** $\pm$0.006 |
| Other Type I Bridge | 0.728 $\pm$0.007 |
| Type II Bridge | **0.765** $\pm$0.003 |
| Other Type II Bridge | 0.732 $\pm$0.005 |
| Other Bezier | 0.712 $\pm$0.003 |



model: ResNet-32×2 for CIFAR-10, ResNet-32×4 for CIFAR-100, ResNet-18 for Tiny ImageNet and ResNet-50 for ImageNet, where ×2 and ×4 respectively denotes doubling and quadrupling of the number of channels for convolutional layers. We construct bridge networks with CNNs with a residual path whose inference costs are relatively low compared to those of ResNet base models. For detailed training settings, including bridge network architectures or hyperparameter settings, please refer to Appendix A.2.

By changing the channel size of the convolutional layers in the bridge network, we can balance the trade-off between performance gains with computational costs. We check this trade-off in Table 2. We refer to a bridge network with less than 15% of floating-point operations (FLOPs) compared to the base model as Bridge$_{sm}$(small version), and a bridge with more than 15% as Bridge$_{md}$(medium version).

**Efficiency metrics**  We choose FLOPs and the number of parameters (#Params) for efficiency evaluation as these metrics are commonly used to consider the efficiency (Dehghani et al., 2021). Because FLOPs and #Params of the base model are different for each dataset, we report the relative FLOPs and the relative #Params with respect to the corresponding base model instead for better comparison.

**Uncertainty metrics**  As suggested by Ashukha et al. (2020), along with the classification accuracy (ACC), we report the calibrated versions of Negative Log-likelihood (NLL), Expected Calibration Error (ECE), and Brier Score (BS) as metrics for uncertainty evaluation. We also measure the Deep Ensemble Equivalent (DEE) score proposed in Ashukha et al. (2020), which shows the relative performance for DE in terms of NLL and roughly be interpreted as *effective number of models* for an ensemble. See Appendix A.3 for more details.

**Table 2:** FLOPs, #Params, $R^2$ scores, and *ensemble* performance metrics of various type II bridge network sizes on CIFAR-100. We use ResNet-32×4 as a base model and 3 blocks of CNN with a residual connection as bridge networks. The number after CNN indicates the number of channels. $R^2$ scores are measured with respect to the target Bezier $r = 0.5$.

| Bridge | FLOPs ($\downarrow$) | #Params ($\downarrow$) | $R^2$ ($\uparrow$) | ACC ($\uparrow$) | NLL ($\downarrow$) | ECE ($\downarrow$) | BS ($\downarrow$) |
|---|---|---|---|---|---|---|---|
| CNN 32 ch | × 0.012 | × 0.009 | 0.709 ± 0.004 | 75.62 ± 0.17 | 0.914 ± 0.005 | **0.013** ± 0.001 | 0.342 ± 0.002 |
| CNN 64 ch | × 0.029 | × 0.022 | 0.758 ± 0.004 | 75.78 ± 0.30 | 0.901 ± 0.004 | 0.016 ± 0.002 | 0.338 ± 0.001 |
| CNN 128 ch | × 0.079 | × 0.060 | 0.793 ± 0.003 | 75.98 ± 0.20 | 0.894 ± 0.003 | 0.021 ± 0.002 | 0.335 ± 0.001 |
| CNN 256 ch | × 0.246 | × 0.188 | **0.814** ± 0.002 | **76.13** ± 0.14 | **0.890** ± 0.004 | 0.023 ± 0.003 | **0.334** ± 0.002 |

**Table 3:** Performance improvement of the ensemble by adding type I bridges to the single base ResNet model on Tiny ImageNet dataset. FLOPs, #Params, and DEE metrics are measured with respect to the single base model. Bridge$_{sm}$ and Bridge$_{md}$ denote the small and the medium versions of the bridge network based on their FLOPs.

| Model | FLOPs ($\downarrow$) | #Params ($\downarrow$) | ACC ($\uparrow$) | NLL ($\downarrow$) | ECE ($\downarrow$) | BS ($\downarrow$) | DEE ($\uparrow$) |
|---|---|---|---|---|---|---|---|
| ResNet (DE-1) | × 1.000 | × 1.000 | 63.42 ± 0.23 | 1.618 ± 0.005 | 0.037 ± 0.002 | 0.485 ± 0.003 | 1.000 |
| + 1 Bridge$_{sm}$ | × 1.088 | × 1.093 | 65.38 ± 0.09 | 1.444 ± 0.005 | 0.015 ± 0.001 | 0.461 ± 0.001 | 2.179 ± 0.110 |
| + 2 Bridge$_{sm}$ | × 1.176 | × 1.186 | 65.55 ± 0.15 | 1.405 ± 0.005 | **0.013** ± 0.001 | 0.456 ± 0.001 | 2.750 ± 0.086 |
| + 3 Bridge$_{sm}$ | × 1.264 | × 1.279 | **65.61** ± 0.10 | **1.388** ± 0.003 | 0.014 ± 0.002 | **0.455** ± 0.000 | **3.022** ± 0.079 |
| + 1 Bridge$_{md}$ | × 1.277 | × 1.290 | 65.94 ± 0.15 | 1.418 ± 0.003 | 0.018 ± 0.002 | 0.453 ± 0.001 | 2.562 ± 0.056 |
| + 2 Bridge$_{md}$ | × 1.554 | × 1.580 | 66.59 ± 0.09 | 1.372 ± 0.001 | 0.016 ± 0.002 | 0.445 ± 0.000 | 3.437 ± 0.036 |
| + 3 Bridge$_{md}$ | × 1.831 | × 1.870 | **66.79** ± 0.11 | **1.353** ± 0.001 | **0.015** ± 0.001 | **0.443** ± 0.000 | **3.967** ± 0.043 |
| DE-2 | × 2.000 | × 2.000 | 66.21 ± 0.10 | 1.456 ± 0.004 | 0.022 ± 0.002 | 0.450 ± 0.002 | 2.000 |

## 5.2 CORRESPONDENCE BETWEEN BRIDGE NETWORK AND BEZIER CURVE

To assess the quality of the prediction of bridge networks, we use a set of ensemble parameters $\{\boldsymbol{\theta}_1, \boldsymbol{\theta}_2, \ldots, \boldsymbol{\theta}_m\}$ and Bezier curves between them. If the bridge network $H_{1,2}^{(0.5)}$ predicts $\boldsymbol{v}_{1,2}(0.5)$ well compared to the other baselines, we can confirm that there exists the correspondence between the bridge network and the Bezier curve. To this end, we measure the $R^2$ score which quantifies how similar outputs of the following baselines to that of the target function $f_{\boldsymbol{\theta}_{1,2}^{(be)}(0.5)}$; (1) 'Type I/II Bridge' denote the bridge network imitating the function of $\boldsymbol{\theta}_{1,2}^{(be)}(0.5)$, (2) 'Other Type I/II Bridge' denote the bridge network imitating the function of $\boldsymbol{\theta}_{i,j}^{(be)}(0.5)$ for some $(i, j) \neq (1, 2)$, and (3) 'Other Bezier' denotes the base model with the parameters $\boldsymbol{\theta}_{i,j}^{(be)}(0.5)$ for some $(i, j) \neq (1, 2)$.

Table 1 summarizes the results. Compared to the baselines (i.e., 'Other Type I/II Bridge' and 'Other Bezier'), the bridge networks produce more similar outputs to the target outputs. The $R^2$ values between the predictions and targets are significantly higher than those from the wrong targets, demonstrating that the bridge predictions indeed are approximating our target outputs of interest.

Fig. 3 visualizes whether the bridge network $H_{1,2}^{(0.5)}$ predicts the logits from $\boldsymbol{\theta}_{1,2}^{(be)}(0.5)$. To be more specific, we visualize the predicted logits from $\boldsymbol{\theta}_1$, $\boldsymbol{\theta}_2$, $\boldsymbol{\theta}_{1,2}^{(be)}(0.5)$, and the bridge network $H_{1,2}^{(0.5)}$, for two test examples of CIFAR-10. Indeed, the bridge network predicts well the logits from the Bezier parameter. Appendix B.1 provides additional examples which further verify this.

**Relation between model size and regression result** We measure the relation between the size of bridge networks and the goodness of fits of prediction measured by $R^2$ scores. Table 2 shows that we can achieve decent $R^2$ scores with a small number of parameters, and the prediction gets better as we increase the flexibility of our bridge network. Also, the results show that a higher $R^2$ score leads to better ensemble results.

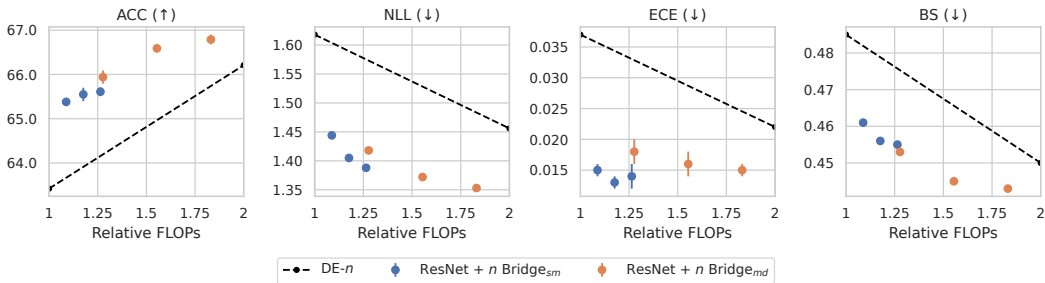

**Figure 4:** The cost-performance plots of *type I bridge(s)* compared to DE on Tiny ImageNet. The x-axis denotes the relative FLOPs quantifying the inference cost of the model compared to a single base model, and the y-axis shows the corresponding predictive performance. On the basis of DE (black dashed line), the upper left position is preferable in ACC, and the lower left position is preferable in NLL, ECE, and BS.

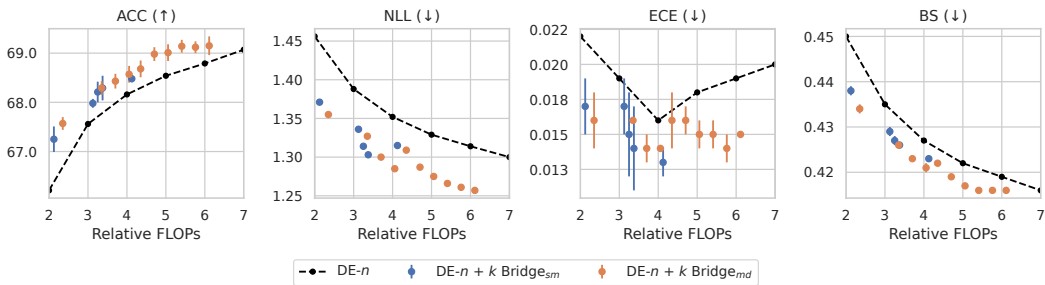

**Figure 5:** The cost-performance plots of *type II bridge(s)* compared to DE on Tiny ImageNet. Others are identical to Fig. 4 except that we extend the DE basis from DE-2 to DE-7 (black dashed lines).

## 5.3 CLASSIFICATION WITH BRIDGE NETWORKS

### 5.3.1 TYPE I BRIDGE NETWORKS

**Single Model performances with type I bridge networks**  In situations where multiple forward passes are not allowed for inference, we can approximate an ensemble of a single base model and the ones from Bezier curves with type I bridge networks. The results are shown in Table 3. The results show that for Stochastic Gradient Descent (SGD) trained single base ResNet model, an ensemble with type I bridge networks improves the performance both in terms of accuracy and uncertainty estimation. Only adding one small type I bridge to the base model (ResNet + 1 Bridge$_{sm}$) dramatically improves the accuracy ($\approx \times 1.701$) and DEE ($\approx \times 2.179$).

**Using multiple type I bridge networks**  As type I bridge network requires features from only one mode of each curve for inference, we can use multiple type I bridge networks for a single base model without significantly increasing inference cost, as we mentioned at Eq. 8. Table 3 reports the performance gain of a single base model with increasing number of type I bridges. Each bridge approximates the models on different Bezier curves between a single mode and others (i.e., Bezier curves between modes A-B, A-C, and so on where A, B, and C are different modes.), not the models on a single Bezier curve. Adding more bridge networks introduces more diverse outputs to the ensembles. One can see that the performance continuously improves as the number of bridges increases, with low additional inference cost. Fig. 4 shows how much type I bridge networks efficiently increase the performances proportional to FLOPs.

### 5.3.2 TYPE II BRIDGE NETWORKS

**Performance**  Table 4 summarizes the classification results comparing DE, DE with Bezier curves, and DE with type II bridge networks. For the more experimental results including other datasets,

**Table 4:** Performance improvement of the ensemble by adding type II bridges as members to existing DE ensembles on Tiny ImageNet dataset. FLOPs, #Params, and DEE metrics are measured with respect to corresponding DEs. Type II bridges consistently improve the accuracy and uncertainty metrics of the ensemble before saturation. Bridge$_{sm}$ and Bridge$_{md}$ denote the small and the medium versions of the bridge network based on their FLOPs.

| Model | FLOPs ($\downarrow$) | #Params ($\downarrow$) | ACC ($\uparrow$) | NLL ($\downarrow$) | ECE ($\downarrow$) | BS ($\downarrow$) | DEE ($\uparrow$) |
|---|---|---|---|---|---|---|---|
| DE-4 | $\times$ 4.000 | $\times$ 4.000 | $68.16 \pm 0.11$ | $1.352 \pm 0.001$ | $0.016 \pm 0.000$ | $0.427 \pm 0.001$ | 4.000 |
| + 1 Bridge$_{sm}$ | $\times$ 4.125 | $\times$ 4.132 | $68.48 \pm 0.07$ | $1.315 \pm 0.002$ | $0.013 \pm 0.001$ | $0.423 \pm 0.000$ | $5.962 \pm 0.127$ |
| + 2 Bridge$_{sm}$ | $\times$ 4.250 | $\times$ 4.264 | $68.67 \pm 0.10$ | $1.297 \pm 0.002$ | $0.015 \pm 0.001$ | $0.422 \pm 0.000$ | $7.239 \pm 0.226$ |
| + 4 Bridge$_{sm}$ | $\times$ 4.500 | $\times$ 4.528 | $\mathbf{68.69} \pm 0.18$ | $1.281 \pm 0.002$ | $0.012 \pm 0.002$ | $\mathbf{0.421} \pm 0.000$ | $8.432 \pm 0.383$ |
| + 6 Bridge$_{sm}$ | $\times$ 4.750 | $\times$ 4.792 | $68.58 \pm 0.04$ | $\mathbf{1.276} \pm 0.002$ | $\mathbf{0.011} \pm 0.001$ | $0.422 \pm 0.000$ | $\mathbf{8.768} \pm 0.441$ |
| + 1 Bridge$_{md}$ | $\times$ 4.352 | $\times$ 4.367 | $68.68 \pm 0.17$ | $1.309 \pm 0.002$ | $0.016 \pm 0.002$ | $0.422 \pm 0.000$ | $6.377 \pm 0.178$ |
| + 2 Bridge$_{md}$ | $\times$ 4.704 | $\times$ 4.734 | $68.98 \pm 0.14$ | $1.287 \pm 0.002$ | $0.016 \pm 0.001$ | $0.419 \pm 0.000$ | $7.986 \pm 0.347$ |
| + 3 Bridge$_{md}$ | $\times$ 5.056 | $\times$ 5.101 | $69.01 \pm 0.17$ | $1.275 \pm 0.001$ | $\mathbf{0.015} \pm 0.001$ | $0.417 \pm 0.000$ | $8.892 \pm 0.389$ |
| + 4 Bridge$_{md}$ | $\times$ 5.408 | $\times$ 5.468 | $\mathbf{69.14} \pm 0.13$ | $\mathbf{1.266} \pm 0.001$ | $\mathbf{0.015} \pm 0.001$ | $\mathbf{0.416} \pm 0.000$ | $\mathbf{9.539} \pm 0.481$ |
| DE-5 | $\times$ 5.000 | $\times$ 5.000 | $68.54 \pm 0.08$ | $1.329 \pm 0.001$ | $0.018 \pm 0.001$ | $0.422 \pm 0.001$ | 5.000 |

please refer to Appendix B. From Table 4, one can see that with only sightly increase in the computational costs, the ensembles with bridge networks achieves almost DEE 1.962 ensemble gain for DE-4 case. This gain is not specific only for DE-4; the ensembles with type II bridge networks consistently improved predictive accuracy and uncertainty calibration with negligible increase in the inference costs. Fig. 5 shows how much our type II bridge network achieve high performance in the perspective of relative FLOPs.

**Computational cost** We report FLOPs for inference on Table 4 to indicate how much relative computational costs are required for the competing models. Fig. 5 summarizes the tradeoff between FLOPs and performance in various metrics. As one can see from these results, our bridge networks could achieve remarkable gain in performance, so for some cases, adding bridge ensembles achieved performance gains larger than those might be achieved by adding entire ensemble members. For instance, in Tiny ImageNet experiments, DE-4 + 2 bridges was better than DE-5 (DEE $\approx \times 7.239$). Please refer to Appendix B for the full results including various DE size and other datasets.

## 5.4 How many type II bridges are required?

For an ensemble of $m$ parameters, the number of pairs can be connected by Bezier curves is $\binom{m}{2}$, which grows quadratically with $m$. In the previous experiment, we constructed Bezier curves and bridges for all possible pairs (which explains the large inference costs for Bezier ensembles), but in practice, we found that it is not necessary to use bridge networks for all of those pairs. As an example, we compare the performance of DE-4 + bridge ensembles with increasing number of bridges on Tiny ImageNet dataset. The results are summarized in Table 4. Just one bridge dramatically increases the performance, and the performance gain gradually saturates as we add more bridges. Notably, only one bridge would suffice to outperform DE-5 (DEE $\approx \times 5.962$).

## 6 Conclusion

In this paper, we proposed a novel framework for efficient ensembling that reduces inference costs of ensembles with a lightweight network called bridge networks. Bridge networks predict the neural network outputs corresponding to the parameters obtained from the Bezier curves connecting two ensemble parameters without actual forward passes through the network. Instead, they reuse features and outputs computed from the ensemble members and predict the outputs corresponding to Bezier parameters directly in function spaces. Using various image classification benchmarks, we demonstrate that we can train such bridge networks with simple CNNs with minimal inference costs, and bridge augmented ensembles could achieve significant gain both in terms of accuracy and uncertainty calibration.

**Reproducibility statement** Please refer to Appendix A for full experimental detail including datasets, models, evaluation metrics and computing resources.

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

## A EXPERIMENTAL DETAILS

### A.1 FILTER RESPONSE NORMALIZATION

Throughout experiments using convolutional neural networks, we use the Filter Response Normalization (FRN; Singh and Krishnan, 2020) instead of the Batch Normalization (BN; Ioffe and Szegedy, 2015) to avoid recomputation of BN statistics along the subspaces. Besides, FRN is fully made up of learned parameters and it does not utilize dependencies between training examples, thus, it gives us a more clear interpretation of the parameter space (Wenzel et al., 2020; Izmailov et al., 2021).

### A.2 DATASETS AND MODELS

**Dataset** We use CIFAR-10, CIFAR-100 (Krizhevsky et al., 2009), Tiny ImageNet (Li et al., 2017) and ImageNet (Russakovsky et al., 2015) datasets. We apply the data augmentation consisting of random cropping of 32 pixels with padding of 4 pixels and random horizontal flipping. We subtract per-channel means from input images and divide them by per-channel standard deviations.

**Network** We use CNN with residual path similar to the ResNet block structure. To use the features of base models, we embed one or more features from different layers of base models.

For CIFAR-10 dataset, we use ResNet-32×2 as a base network which consists of 15 blocks and 32 layers with widen factor of 2, and we use CNN 3 blocks as type I and type II bridge networks. The bridge networks use the features $z$ of the third to last block.

For CIFAR-100 dataset, we use ResNet-32×4 as a base network which is almost same as ResNet-32 with widen factor of 2, and we use CNN 3 blocks as type I and type II bridge networks. The bridge networks use the features $z$ of the third to last block.

For Tiny ImageNet dataset, we use ResNet-18 as a base network which consists of 8 blocks and 18 layers, and we use CNN 2 blocks as a type I and type II bridge network. The bridge networks use the features $z$ of the third to last and the second to last blocks.

For ImageNet dataset, we use ResNet-50 as a base network which consists of 17 blocks and 50 layers, and we use CNN 3 blocks as a type I and type II bridge network. The bridge networks use the features $z$ of the third to last and the second to last blocks.

**Optimization** We train base ResNet networks for 200 epochs with learning rate 0.1. We use the SGD optimizer with momentum 0.9 and adjust learning rate with simple cosine scheduler. We give weight decay 0.001 for CIFAR-10 dataset, 0.0005 for CIFAR-100 and Tiny ImageNet dataset, and 0.0001 for ImageNet dataset.

**Regularization** We introduced two additional hyperparameters for training bridge models; 1) the regularization scale $\lambda$ and 2) the mixup coefficient $\alpha$. Since the training error of the base network is near zero for the family of residual networks on CIFAR-10/100, given a training input without any modification, the base network and the target network (the one on the Bezier curve) will produce almost identical outputs, so the bridge trained with them will just copy the outputs of the base network. To prevent this, we perturb the inputs via mixup, and regularize the bridge to produce outputs different from the ones computed from the base models. On the other hand, for the datasets such as ImageNet where the models fail to achieve near zero training errors, the base network and the target networks are already distinct enough, so we found that the bridge can be trained easily without such tricks (i.e., we used $\lambda = 0.0$ and $\alpha = 0.0$). We search $0.0 \leq \lambda \leq 0.4$ for the regularization scale $\lambda$. We use $\alpha = 0.4$ for CIFAR-10/100 and Tiny ImageNet datasets. We do not use mixup($\alpha = 0.0$) for ImageNet dataset.

### A.3 EVALUATION

**Efficiency metrics**   Dehghani et al. (2021) pointed out that there can be contradictions between commonly used metrics (e.g., FLOPs, the number of parameters, and speed) and suggested refraining from reporting results using just a single one. So, we present FLOPs and the number of parameters in the results.

**Uncertainty metrics**   Let $p(x) \in [0, 1]^K$ be a predicted probabilities for a given input $x$, where $p^{(k)}$ denotes the $k$th element of the probability vector, i.e., $p^{(k)}$ is a predicted confidence on $k$th class. We have the following common metrics on the dataset $\mathcal{D}$ consists of inputs $x$ and labels $y$:

- Accuracy (ACC):

$$\mathrm{ACC}(\mathcal{D}) = \mathbb{E}_{(x,y)\in\mathcal{D}}\left[\left[y = \arg\max_k p^{(k)}(x)\right]\right]. \tag{11}$$

- Negative log-likelihood (NLL):

$$\mathrm{NLL}(\mathcal{D}) = \mathbb{E}_{(x,y)\in\mathcal{D}}\left[-\log p^{(y)}(x)\right]. \tag{12}$$

- Brier score (BS):

$$\mathrm{BS}(\mathcal{D}) = \mathbb{E}_{(x,y)\in\mathcal{D}}\left[\left\|p(x) - y\right\|_2^2\right], \tag{13}$$

  where $y$ denotes one-hot encoded version of the label $y$, i.e., $y^{(y)} = 1$ and $y^{(k)} = 0$ for $k \neq y$.

- Expected calibration error (ECE):

$$\mathrm{ECE}(\mathcal{D}, N_{\mathrm{bin}}) = \sum_{b=1}^{N_{\mathrm{bin}}} \frac{n_b|\delta_b|}{n_1 + \cdots + n_{N_{\mathrm{bin}}}}, \tag{14}$$

  where $N_{\mathrm{bin}}$ is the number of bins, $n_b$ is the number of examples in the $b$th bin, and $\delta_b$ is the calibration error of the $b$th bin. Specifically, the $b$th bin consists of predictions having the maximum confidence values in $[(b-1)/K, b/K)$, and the calibration error denotes the difference between accuracy and averaged confidences. We fix $N_{\mathrm{bin}} = 15$ in this paper.

We evaluate the *calibrated* metrics that compute the aforementioned metrics with the temperature scaling (Guo et al., 2017), as Ashukha et al. (2020) suggested. Specifically, (1) we first find the optimal temperature which minimizes the NLL over the validation examples, and (2) compute uncertainty metrics including NLL, BS, and ECE using temperature scaled predicted probabilities under the optimal temperature. Moreover, we evaluate the following Deep Ensemble Equivalent (DEE) score, which measure the relative performance for DE in terms of NLL,

$$\mathrm{DEE}(\mathcal{D}) = \min\{m \geq 0 \mid \mathrm{NLL}(\mathcal{D}) \leq \mathrm{NLL}_{\mathrm{DE}\text{-}m}(\mathcal{D})\}, \tag{15}$$

where $\mathrm{NLL}_{\mathrm{DE}\text{-}m}(\mathcal{D})$ denotes the NLL of DE-$m$ on the dataset $\mathcal{D}$. Here, we linearly interpolate $\mathrm{NLL}_{\mathrm{DE}\text{-}m}(\mathcal{D})$ values for $m \in \mathbb{R}$ and make the DEE score continuous.

## A.4   COMPUTING RESOURCES

We conduct Tiny ImageNet experiments on 8 TPUv2 and 8 TPUv3 cores, supported by TPU Research Cloud[1] and the others on 8 RTX3090 cores. We attached code to the supplimentary material. We use PyTorch (Paszke et al., 2019) with BSD-style license. Visit PyTorch GitHub repository[2] for more details.

---

[1]https://sites.research.google/trc/about/
[2]https://github.com/pytorch/pytorch/blob/master/LICENSE

# B ADDITIONAL EXPERIMENTS

## B.1 ADDITIONAL EXAMPLES

We visually inspect the logit regression of type II bridge network. Our bridge network very accurately predicts the logits of $r = 0.5$ from Bezier curve when the two base models ($r = 0$ and $r = 1$) gives similar output logits (deer, ship, and frog). When the base models are not confident on the samples (airplane, bird, cat and horse), the network recovers the scale of logits approximately. However it fails to predict some very difficult samples (truck and dog) when even the base models are very confused.

## B.2 FULL TYPE I AND TYPE II BRIDGE RESULTS

We report full experimental results for classification tasks; 1) Type I bridge network results in Table 5, Table 7, Table 9, and Table 11, 2) Type II bridge network results in Table 6, Table 8, Table 10 and Table 12.

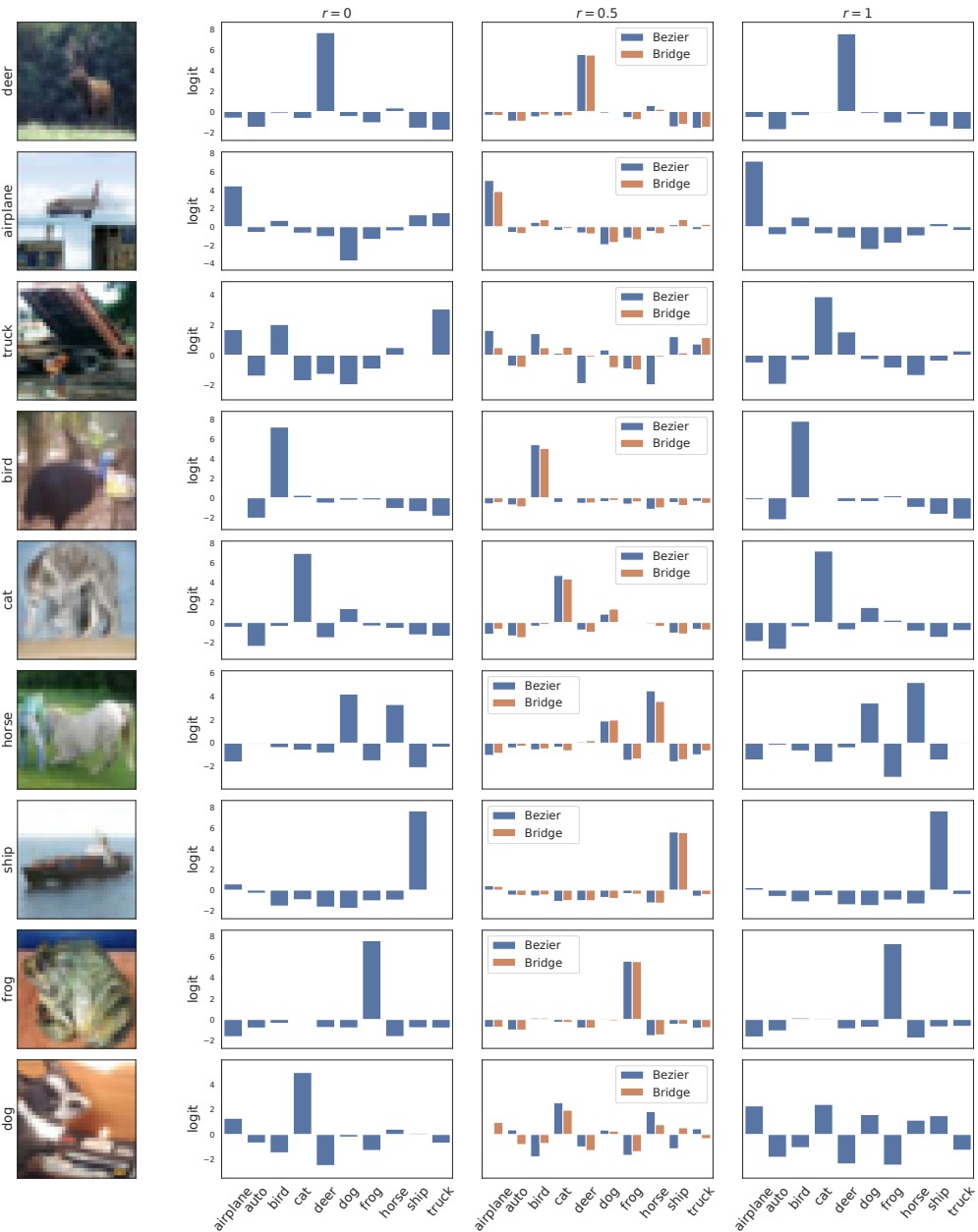

**Figure 6:** Bar plots in the third column depict whether the bridge network (orange) outputs the same logit values as the base model with the Bezier parameters $\boldsymbol{\theta}_{1,2}^{(\mathrm{be})}(0.5)$ (blue), for a given test inputs displayed in the first column. We also depicts the predicted logits from $\boldsymbol{\theta}_1$ and $\boldsymbol{\theta}_2$ in the second and fourth columns, respectively.

**Table 5:** Full result of performance improvement of the ensemble by adding type I bridges on CIFAR-10 dataset. We use same settings as described in Table 3.

| Model | FLOPs (↓) | #Params (↓) | ACC (↑) | NLL (↓) | ECE (↓) | BS (↓) | DEE (↑) |
|---|---|---|---|---|---|---|---|
| ResNet (DE-1) | × 1.000 | × 1.000 | 91.78 ± 0.10 | 0.287 ± 0.001 | 0.019 ± 0.001 | 0.126 ± 0.001 | 1.000 |
| + 1 Bridge$_{sm}$ | × 1.062 | × 1.048 | 92.07 ± 0.04 | 0.254 ± 0.001 | 0.009 ± 0.001 | 0.119 ± 0.000 | 1.606 ± 0.035 |
| + 2 Bridge$_{sm}$ | × 1.124 | × 1.096 | **92.13** ± 0.06 | **0.250** ± 0.001 | **0.008** ± 0.001 | **0.118** ± 0.000 | 1.677 ± 0.033 |
| + 3 Bridge$_{sm}$ | × 1.186 | × 1.144 | **92.13** ± 0.04 | **0.250** ± 0.000 | 0.009 ± 0.000 | **0.118** ± 0.000 | 1.692 ± 0.033 |
| + 4 Bridge$_{sm}$ | × 1.248 | × 1.192 | 92.12 ± 0.03 | **0.250** ± 0.000 | 0.009 ± 0.001 | 0.119 ± 0.000 | **1.695** ± 0.041 |
| + 1 Bridge$_{md}$ | × 1.205 | × 1.159 | 92.09 ± 0.05 | 0.253 ± 0.001 | 0.009 ± 0.000 | 0.119 ± 0.000 | 1.623 ± 0.034 |
| + 2 Bridge$_{md}$ | × 1.410 | × 1.318 | **92.17** ± 0.06 | 0.249 ± 0.001 | 0.009 ± 0.000 | **0.118** ± 0.000 | 1.705 ± 0.047 |
| + 3 Bridge$_{md}$ | × 1.615 | × 1.477 | 92.15 ± 0.04 | 0.248 ± 0.001 | **0.008** ± 0.001 | **0.118** ± 0.000 | 1.729 ± 0.048 |
| + 4 Bridge$_{md}$ | × 1.820 | × 1.636 | 92.14 ± 0.05 | **0.247** ± 0.001 | 0.009 ± 0.001 | **0.118** ± 0.000 | **1.736** ± 0.049 |
| DE-2 | × 2.000 | × 2.000 | 93.07 ± 0.11 | 0.233 ± 0.003 | 0.012 ± 0.001 | 0.107 ± 0.001 | 2.000 |

**Table 6:** Full result of performance improvement of the ensemble by adding type II bridges on CIFAR-10 dataset. We use same settings as described in Table 4.

| Model | FLOPs (↓) | #Params (↓) | ACC (↑) | NLL (↓) | ECE (↓) | BS (↓) | DEE (↑) |
|---|---|---|---|---|---|---|---|
| DE-2 | × 2.000 | × 2.000 | 93.07 ± 0.11 | 0.233 ± 0.003 | 0.012 ± 0.001 | 0.107 ± 0.001 | 2.000 |
| + 1 Bridge$_{sm}$ | × 2.081 | × 2.063 | **93.09** ± 0.09 | **0.220** ± 0.003 | **0.008** ± 0.001 | **0.104** ± 0.001 | **2.774** ± 0.084 |
| + 1 Bridge$_{md}$ | × 2.247 | × 2.192 | **93.07** ± 0.07 | **0.219** ± 0.003 | **0.006** ± 0.001 | **0.103** ± 0.001 | **2.808** ± 0.105 |
| + 1 Bezier | × 3.000 | × 3.000 | 93.09 ± 0.10 | 0.227 ± 0.004 | 0.009 ± 0.000 | 0.104 ± 0.001 | 2.357 ± 0.080 |
| DE-3 | × 3.000 | × 3.000 | 93.37 ± 0.03 | 0.216 ± 0.004 | 0.009 ± 0.001 | 0.100 ± 0.001 | 3.000 |
| + 1 Bridge$_{sm}$ | × 3.081 | × 3.063 | 93.46 ± 0.06 | 0.206 ± 0.002 | 0.007 ± 0.002 | 0.098 ± 0.001 | 3.880 ± 0.118 |
| + 2 Bridge$_{sm}$ | × 3.162 | × 3.126 | **93.50** ± 0.04 | 0.204 ± 0.002 | **0.006** ± 0.001 | **0.097** ± 0.001 | 4.214 ± 0.143 |
| + 3 Bridge$_{sm}$ | × 3.243 | × 3.189 | **93.50** ± 0.10 | **0.203** ± 0.002 | **0.006** ± 0.001 | **0.097** ± 0.001 | **4.384** ± 0.178 |
| + 1 Bridge$_{md}$ | × 3.247 | × 3.192 | 93.45 ± 0.04 | 0.206 ± 0.002 | 0.007 ± 0.001 | **0.097** ± 0.001 | 3.918 ± 0.117 |
| + 2 Bridge$_{md}$ | × 3.494 | × 3.384 | **93.54** ± 0.02 | 0.203 ± 0.002 | **0.006** ± 0.002 | **0.097** ± 0.001 | 4.301 ± 0.157 |
| + 3 Bridge$_{md}$ | × 3.741 | × 3.576 | 93.53 ± 0.08 | **0.202** ± 0.002 | **0.006** ± 0.001 | **0.097** ± 0.001 | **4.539** ± 0.171 |
| + 3 Bezier | × 6.000 | × 6.000 | 93.44 ± 0.10 | 0.207 ± 0.003 | 0.008 ± 0.001 | 0.097 ± 0.001 | 3.852 ± 0.214 |
| DE-4 | × 4.000 | × 4.000 | 93.59 ± 0.10 | 0.205 ± 0.002 | 0.010 ± 0.001 | 0.096 ± 0.001 | 4.000 |
| + 1 Bridge$_{sm}$ | × 4.081 | × 4.063 | 93.58 ± 0.06 | 0.199 ± 0.002 | 0.007 ± 0.001 | **0.094** ± 0.001 | 5.017 ± 0.088 |
| + 2 Bridge$_{sm}$ | × 4.162 | × 4.126 | **93.64** ± 0.03 | 0.197 ± 0.002 | 0.006 ± 0.001 | **0.094** ± 0.001 | 5.333 ± 0.044 |
| + 3 Bridge$_{sm}$ | × 4.243 | × 4.189 | 93.58 ± 0.04 | **0.196** ± 0.002 | 0.006 ± 0.002 | **0.094** ± 0.001 | 5.484 ± 0.090 |
| + 4 Bridge$_{sm}$ | × 4.324 | × 4.252 | **93.64** ± 0.06 | **0.196** ± 0.002 | 0.007 ± 0.001 | **0.094** ± 0.001 | 5.590 ± 0.123 |
| + 5 Bridge$_{sm}$ | × 4.405 | × 4.315 | 93.58 ± 0.04 | **0.196** ± 0.002 | **0.005** ± 0.001 | **0.094** ± 0.001 | **5.647** ± 0.165 |
| + 6 Bridge$_{sm}$ | × 4.486 | × 4.378 | 93.60 ± 0.10 | **0.196** ± 0.002 | **0.005** ± 0.000 | **0.094** ± 0.001 | 5.607 ± 0.120 |
| + 1 Bridge$_{md}$ | × 4.247 | × 4.192 | 93.57 ± 0.04 | 0.198 ± 0.002 | 0.007 ± 0.000 | **0.094** ± 0.001 | 5.055 ± 0.064 |
| + 2 Bridge$_{md}$ | × 4.494 | × 4.384 | 93.61 ± 0.05 | 0.197 ± 0.002 | 0.006 ± 0.001 | **0.094** ± 0.001 | 5.428 ± 0.069 |
| + 3 Bridge$_{md}$ | × 4.741 | × 4.576 | **93.64** ± 0.09 | 0.196 ± 0.002 | 0.006 ± 0.001 | **0.094** ± 0.001 | 5.610 ± 0.141 |
| + 4 Bridge$_{md}$ | × 4.988 | × 4.768 | **93.64** ± 0.10 | **0.195** ± 0.002 | **0.005** ± 0.001 | **0.094** ± 0.001 | 5.774 ± 0.183 |
| + 5 Bridge$_{md}$ | × 5.235 | × 4.960 | 93.58 ± 0.08 | **0.195** ± 0.002 | **0.005** ± 0.001 | **0.094** ± 0.001 | 5.849 ± 0.156 |
| + 6 Bridge$_{md}$ | × 5.482 | × 5.152 | 93.62 ± 0.05 | **0.195** ± 0.002 | **0.005** ± 0.000 | **0.094** ± 0.001 | **5.855** ± 0.114 |
| + 6 Bezier | × 10.000 | × 0.000 | 93.63 ± 0.10 | 0.197 ± 0.002 | 0.005 ± 0.001 | 0.094 ± 0.001 | 5.322 ± 0.151 |
| DE-5 | × 5.000 | × 5.000 | 93.68 ± 0.10 | 0.199 ± 0.002 | 0.011 ± 0.001 | 0.093 ± 0.001 | 5.000 |

**Table 7:** Full result of performance improvement of the ensemble by adding type I bridges on CIFAR-100 dataset. We use same settings as described in Table 3.

| Model | FLOPs (↓) | #Params (↓) | ACC (↑) | NLL (↓) | ECE (↓) | BS (↓) | DEE (↑) |
|---|---|---|---|---|---|---|---|
| ResNet (DE-1) | × 1.000 | × 1.000 | 73.11 ± 0.10 | 1.094 ± 0.008 | 0.047 ± 0.003 | 0.379 ± 0.002 | 1.000 |
| + 1 Bridge$_{sm}$ | × 1.062 | × 1.047 | 73.94 ± 0.16 | 0.981 ± 0.004 | 0.022 ± 0.003 | 0.363 ± 0.001 | 1.802 ± 0.036 |
| + 2 Bridge$_{sm}$ | × 1.124 | × 1.094 | 74.02 ± 0.05 | 0.957 ± 0.004 | 0.018 ± 0.002 | **0.360** ± 0.001 | 1.984 ± 0.064 |
| + 3 Bridge$_{sm}$ | × 1.186 | × 1.141 | 74.00 ± 0.09 | 0.948 ± 0.002 | **0.017** ± 0.001 | **0.360** ± 0.001 | 2.078 ± 0.070 |
| + 4 Bridge$_{sm}$ | × 1.248 | × 1.188 | **74.06** ± 0.06 | **0.944** ± 0.001 | 0.018 ± 0.001 | **0.360** ± 0.000 | **2.143** ± 0.062 |
| + 1 Bridge$_{md}$ | × 1.211 | × 1.161 | 74.18 ± 0.11 | 0.965 ± 0.002 | 0.024 ± 0.001 | 0.358 ± 0.001 | 1.914 ± 0.046 |
| + 2 Bridge$_{md}$ | × 1.422 | × 1.322 | 74.41 ± 0.14 | 0.940 ± 0.000 | **0.020** ± 0.001 | 0.353 ± 0.000 | 2.228 ± 0.077 |
| + 3 Bridge$_{md}$ | × 1.633 | × 1.483 | 74.52 ± 0.07 | 0.930 ± 0.001 | **0.020** ± 0.002 | **0.352** ± 0.000 | 2.416 ± 0.055 |
| + 4 Bridge$_{md}$ | × 1.844 | × 1.644 | **74.54** ± 0.02 | **0.925** ± 0.001 | 0.021 ± 0.001 | **0.352** ± 0.000 | **2.517** ± 0.061 |
| DE-2 | × 2.000 | × 2.000 | 75.62 ± 0.17 | 0.952 ± 0.005 | 0.033 ± 0.002 | 0.344 ± 0.002 | 2.000 |

**Table 8:** Full result of performance improvement of the ensemble by adding type II bridges on CIFAR-100 dataset. We use same settings as described in Table 4.

| Model | FLOPs (↓) | #Params (↓) | ACC (↑) | NLL (↓) | ECE (↓) | BS (↓) | DEE (↑) |
|---|---|---|---|---|---|---|---|
| DE-2 | × 2.000 | × 2.000 | 75.62 ± 0.17 | 0.952 ± 0.005 | 0.033 ± 0.002 | 0.344 ± 0.002 | 2.000 |
| + 1 Bridge$_{sm}$ | × 2.079 | × 2.060 | **75.98** ± 0.20 | **0.894** ± 0.003 | **0.021** ± 0.002 | **0.335** ± 0.001 | **3.158** ± 0.061 |
| + 1 Bridge$_{md}$ | × 2.246 | × 2.188 | **76.13** ± 0.14 | **0.890** ± 0.004 | 0.023 ± 0.003 | **0.334** ± 0.002 | **3.290** ± 0.084 |
| + 1 Bezier | × 3.000 | × 3.000 | 76.21 ± 0.10 | 0.909 ± 0.002 | 0.021 ± 0.003 | 0.334 ± 0.001 | 2.818 ± 0.034 |
| DE-3 | × 3.000 | × 3.000 | 76.58 ± 0.11 | 0.899 ± 0.003 | 0.025 ± 0.003 | 0.329 ± 0.001 | 3.000 |
| + 1 Bridge$_{sm}$ | × 3.079 | × 3.060 | 76.74 ± 0.09 | 0.859 ± 0.003 | 0.019 ± 0.002 | 0.325 ± 0.001 | 4.357 ± 0.107 |
| + 2 Bridge$_{sm}$ | × 3.158 | × 3.120 | 76.82 ± 0.16 | 0.844 ± 0.002 | **0.016** ± 0.001 | 0.323 ± 0.001 | 5.100 ± 0.109 |
| + 3 Bridge$_{sm}$ | × 3.237 | × 3.180 | **77.04** ± 0.12 | **0.836** ± 0.003 | 0.017 ± 0.002 | **0.322** ± 0.001 | 5.693 ± 0.089 |
| + 1 Bridge$_{md}$ | × 3.246 | × 3.188 | 76.82 ± 0.13 | 0.858 ± 0.003 | 0.020 ± 0.002 | 0.324 ± 0.001 | 4.419 ± 0.095 |
| + 2 Bridge$_{md}$ | × 3.492 | × 3.376 | 77.06 ± 0.08 | 0.841 ± 0.002 | 0.020 ± 0.003 | 0.321 ± 0.001 | 5.333 ± 0.093 |
| + 3 Bridge$_{md}$ | × 3.738 | × 3.564 | **77.09** ± 0.04 | **0.831** ± 0.002 | **0.018** ± 0.002 | **0.320** ± 0.001 | 6.123 ± 0.098 |
| + 3 Bezier | × 6.000 | × 6.000 | 77.37 ± 0.14 | 0.840 ± 0.003 | 0.017 ± 0.001 | 0.317 ± 0.001 | 5.407 ± 0.147 |
| DE-4 | × 4.000 | × 4.000 | 77.14 ± 0.16 | 0.867 ± 0.001 | 0.023 ± 0.001 | 0.321 ± 0.001 | 4.000 |
| + 1 Bridge$_{sm}$ | × 4.079 | × 4.060 | 77.29 ± 0.10 | 0.838 ± 0.003 | 0.019 ± 0.001 | 0.318 ± 0.001 | 5.553 ± 0.116 |
| + 2 Bridge$_{sm}$ | × 4.158 | × 4.120 | 77.35 ± 0.06 | 0.826 ± 0.002 | 0.016 ± 0.001 | **0.317** ± 0.001 | 6.825 ± 0.122 |
| + 3 Bridge$_{sm}$ | × 4.237 | × 4.180 | 77.34 ± 0.06 | 0.820 ± 0.001 | 0.015 ± 0.001 | **0.317** ± 0.001 | 7.675 ± 0.280 |
| + 4 Bridge$_{sm}$ | × 4.316 | × 4.240 | **77.38** ± 0.05 | 0.815 ± 0.002 | 0.016 ± 0.001 | **0.317** ± 0.001 | 8.307 ± 0.331 |
| + 5 Bridge$_{sm}$ | × 4.395 | × 4.300 | 77.33 ± 0.06 | 0.812 ± 0.002 | **0.014** ± 0.002 | **0.317** ± 0.001 | 8.695 ± 0.363 |
| + 6 Bridge$_{sm}$ | × 4.474 | × 4.360 | 77.35 ± 0.02 | **0.811** ± 0.002 | 0.016 ± 0.001 | **0.317** ± 0.001 | **8.862** ± 0.381 |
| + 1 Bridge$_{md}$ | × 4.246 | × 4.188 | 77.40 ± 0.14 | 0.837 ± 0.002 | 0.020 ± 0.001 | 0.317 ± 0.001 | 5.578 ± 0.079 |
| + 2 Bridge$_{md}$ | × 4.492 | × 4.376 | 77.34 ± 0.13 | 0.824 ± 0.001 | **0.016** ± 0.001 | 0.316 ± 0.001 | 7.015 ± 0.156 |
| + 3 Bridge$_{md}$ | × 4.738 | × 4.564 | 77.44 ± 0.11 | 0.817 ± 0.001 | 0.017 ± 0.002 | **0.315** ± 0.001 | 7.987 ± 0.347 |
| + 4 Bridge$_{md}$ | × 4.984 | × 4.752 | 77.41 ± 0.12 | 0.811 ± 0.001 | **0.016** ± 0.001 | **0.315** ± 0.001 | 8.814 ± 0.398 |
| + 5 Bridge$_{md}$ | × 5.230 | × 4.940 | **77.45** ± 0.12 | 0.809 ± 0.002 | 0.018 ± 0.002 | **0.315** ± 0.001 | 9.228 ± 0.529 |
| + 6 Bridge$_{md}$ | × 5.476 | × 5.128 | 77.41 ± 0.14 | **0.806** ± 0.002 | 0.018 ± 0.002 | **0.315** ± 0.001 | **9.563** ± 0.528 |
| + 6 Bezier | × 10.000 | × 10.000 | 77.82 ± 0.06 | 0.808 ± 0.002 | 0.016 ± 0.002 | 0.310 ± 0.001 | 9.270 ± 0.386 |
| DE-5 | × 5.000 | × 5.000 | 77.49 ± 0.12 | 0.845 ± 0.001 | 0.021 ± 0.001 | 0.316 ± 0.000 | 5.000 |

**Table 9:** Full result of performance improvement of the ensemble by adding type I bridges on Tiny ImageNet dataset. We use same settings as described in Table 3.

| Model | FLOPs (↓) | #Params (↓) | ACC (↑) | NLL (↓) | ECE (↓) | BS (↓) | DEE (↑) |
|---|---|---|---|---|---|---|---|
| ResNet (DE-1) | × 1.000 | × 1.000 | 63.42 ± 0.23 | 1.618 ± 0.005 | 0.037 ± 0.002 | 0.485 ± 0.003 | 1.000 |
| + 1 Bridge$_{sm}$ | × 1.088 | × 1.093 | 65.38 ± 0.09 | 1.444 ± 0.005 | 0.015 ± 0.001 | 0.461 ± 0.001 | 2.179 ± 0.110 |
| + 2 Bridge$_{sm}$ | × 1.176 | × 1.186 | 65.55 ± 0.15 | 1.405 ± 0.005 | 0.013 ± 0.001 | 0.456 ± 0.001 | 2.750 ± 0.086 |
| + 3 Bridge$_{sm}$ | × 1.264 | × 1.279 | 65.61 ± 0.10 | 1.388 ± 0.003 | 0.014 ± 0.002 | 0.455 ± 0.000 | 3.022 ± 0.079 |
| + 4 Bridge$_{sm}$ | × 1.352 | × 1.372 | **65.68** ± 0.06 | **1.380** ± 0.002 | **0.012** ± 0.002 | **0.454** ± 0.000 | **3.233** ± 0.084 |
| + 1 Bridge$_{md}$ | × 1.277 | × 1.290 | 65.94 ± 0.15 | 1.418 ± 0.003 | 0.018 ± 0.002 | 0.453 ± 0.001 | 2.562 ± 0.056 |
| + 2 Bridge$_{md}$ | × 1.554 | × 1.580 | 66.59 ± 0.09 | 1.372 ± 0.001 | 0.016 ± 0.002 | 0.445 ± 0.000 | 3.437 ± 0.036 |
| + 3 Bridge$_{md}$ | × 1.831 | × 1.870 | 66.79 ± 0.11 | 1.353 ± 0.001 | **0.015** ± 0.001 | 0.443 ± 0.000 | 3.967 ± 0.043 |
| + 4 Bridge$_{md}$ | × 2.108 | × 2.160 | **66.88** ± 0.15 | **1.342** ± 0.001 | 0.018 ± 0.001 | **0.441** ± 0.000 | **4.450** ± 0.062 |
| DE-2 | × 2.000 | × 2.000 | 66.21 ± 0.10 | 1.456 ± 0.004 | 0.022 ± 0.002 | 0.450 ± 0.002 | 2.000 |

**Table 10:** Full result of performance improvement of the ensemble by adding type II bridges on Tiny ImageNet dataset. We use same settings as described in Table 4.

| Model | FLOPs (↓) | #Params (↓) | ACC (↑) | NLL (↓) | ECE (↓) | BS (↓) | DEE (↑) |
|---|---|---|---|---|---|---|---|
| DE-2 | × 2.000 | × 2.000 | 66.21 ± 0.10 | 1.456 ± 0.004 | 0.022 ± 0.002 | 0.450 ± 0.002 | 2.000 |
| + 1 Bridge$_{sm}$ | × 2.125 | × 2.132 | **67.25** ± 0.26 | **1.371** ± 0.001 | **0.017** ± 0.002 | **0.438** ± 0.001 | **3.478** ± 0.052 |
| + 1 Bridge$_{md}$ | × 2.352 | × 2.367 | **67.57** ± 0.13 | **1.355** ± 0.003 | **0.016** ± 0.002 | **0.434** ± 0.001 | **3.904** ± 0.097 |
| + 1 Bezier | × 3.000 | × 3.000 | 67.43 ± 0.04 | 1.385 ± 0.004 | 0.017 ± 0.002 | 0.436 ± 0.001 | 3.108 ± 0.121 |
| DE-3 | × 3.000 | × 3.000 | 67.56 ± 0.08 | 1.388 ± 0.001 | 0.019 ± 0.001 | 0.435 ± 0.001 | 3.000 |
| + 1 Bridge$_{sm}$ | × 3.125 | × 3.132 | 67.98 ± 0.09 | 1.336 ± 0.002 | 0.017 ± 0.002 | 0.429 ± 0.001 | 4.702 ± 0.079 |
| + 2 Bridge$_{sm}$ | × 3.250 | × 3.264 | 68.21 ± 0.21 | 1.314 ± 0.002 | 0.015 ± 0.003 | 0.427 ± 0.001 | 6.015 ± 0.139 |
| + 3 Bridge$_{sm}$ | × 3.375 | × 3.396 | **68.29** ± 0.25 | **1.303** ± 0.002 | **0.014** ± 0.003 | **0.426** ± 0.000 | **6.822** ± 0.191 |
| + 1 Bridge$_{md}$ | × 3.352 | × 3.367 | 68.29 ± 0.14 | 1.327 ± 0.002 | 0.016 ± 0.000 | 0.426 ± 0.000 | 5.159 ± 0.189 |
| + 2 Bridge$_{md}$ | × 3.704 | × 3.734 | 68.43 ± 0.15 | 1.300 ± 0.002 | **0.014** ± 0.001 | 0.423 ± 0.000 | 7.048 ± 0.250 |
| + 3 Bridge$_{md}$ | × 4.056 | × 4.101 | **68.57** ± 0.17 | **1.285** ± 0.001 | **0.014** ± 0.000 | **0.421** ± 0.001 | **8.110** ± 0.312 |
| + 3 Bezier | × 6.000 | × 6.000 | 68.72 ± 0.21 | 1.307 ± 0.002 | 0.016 ± 0.002 | 0.420 ± 0.001 | 6.464 ± 0.140 |
| DE-4 | × 4.000 | × 4.000 | 68.16 ± 0.11 | 1.352 ± 0.001 | 0.016 ± 0.000 | 0.427 ± 0.001 | 4.000 |
| + 1 Bridge$_{sm}$ | × 4.125 | × 4.132 | 68.48 ± 0.07 | 1.315 ± 0.002 | 0.013 ± 0.001 | 0.423 ± 0.000 | 5.962 ± 0.127 |
| + 2 Bridge$_{sm}$ | × 4.250 | × 4.264 | 68.67 ± 0.10 | 1.297 ± 0.002 | 0.015 ± 0.001 | 0.422 ± 0.000 | 7.239 ± 0.226 |
| + 3 Bridge$_{sm}$ | × 4.375 | × 4.396 | 68.66 ± 0.09 | 1.287 ± 0.001 | 0.012 ± 0.001 | **0.421** ± 0.000 | 7.970 ± 0.298 |
| + 4 Bridge$_{sm}$ | × 4.500 | × 4.528 | **68.69** ± 0.18 | 1.281 ± 0.002 | 0.012 ± 0.002 | **0.421** ± 0.000 | 8.432 ± 0.383 |
| + 5 Bridge$_{sm}$ | × 4.625 | × 4.660 | 68.63 ± 0.11 | 1.279 ± 0.002 | 0.013 ± 0.001 | 0.422 ± 0.000 | 8.596 ± 0.431 |
| + 6 Bridge$_{sm}$ | × 4.750 | × 4.792 | 68.58 ± 0.04 | **1.276** ± 0.002 | **0.011** ± 0.001 | 0.422 ± 0.000 | **8.768** ± 0.441 |
| + 1 Bridge$_{md}$ | × 4.352 | × 4.367 | 68.68 ± 0.17 | 1.309 ± 0.002 | 0.016 ± 0.002 | 0.422 ± 0.000 | 6.377 ± 0.178 |
| + 2 Bridge$_{md}$ | × 4.704 | × 4.734 | 68.98 ± 0.14 | 1.287 ± 0.002 | 0.016 ± 0.001 | 0.419 ± 0.000 | 7.986 ± 0.347 |
| + 3 Bridge$_{md}$ | × 5.056 | × 5.101 | 69.01 ± 0.17 | 1.275 ± 0.002 | 0.015 ± 0.001 | 0.417 ± 0.000 | 8.892 ± 0.389 |
| + 4 Bridge$_{md}$ | × 5.408 | × 5.468 | 69.14 ± 0.13 | 1.266 ± 0.001 | 0.015 ± 0.001 | **0.416** ± 0.000 | 9.539 ± 0.481 |
| + 5 Bridge$_{md}$ | × 5.760 | × 5.835 | 69.12 ± 0.12 | 1.261 ± 0.001 | **0.014** ± 0.001 | **0.416** ± 0.000 | 9.916 ± 0.516 |
| + 6 Bridge$_{md}$ | × 6.112 | × 6.202 | **69.15** ± 0.19 | **1.257** ± 0.000 | 0.015 ± 0.000 | **0.416** ± 0.000 | **10.198** ± 0.493 |
| + 6 Bezier | × 10.000 | × 10.000 | 69.26 ± 0.07 | 1.271 ± 0.002 | 0.016 ± 0.001 | 0.414 ± 0.000 | 9.118 ± 0.383 |
| DE-5 | × 5.000 | × 5.000 | 68.54 ± 0.08 | 1.329 ± 0.001 | 0.018 ± 0.001 | 0.422 ± 0.001 | 5.000 |

**Table 11:** Full result of performance improvement of the ensemble by adding type I bridges on ImageNet dataset. We use same settings as described in Table 3.

| Model | FLOPs ($\downarrow$) | #Params ($\downarrow$) | ACC ($\uparrow$) | NLL ($\downarrow$) | ECE ($\downarrow$) | BS ($\downarrow$) | DEE ($\uparrow$) |
|---|---|---|---|---|---|---|---|
| ResNet (DE-1) | $\times$ 1.000 | $\times$ 1.000 | $75.85 \pm 0.06$ | $1.618 \pm 0.005$ | $0.037 \pm 0.002$ | $0.485 \pm 0.003$ | 1.000 |
| + 1 Bridge$_{md}$ | $\times$ 1.194 | $\times$ 1.222 | $76.57 \pm 0.02$ | $1.418 \pm 0.003$ | $0.018 \pm 0.002$ | $0.453 \pm 0.001$ | $2.562 \pm 0.056$ |
| + 2 Bridge$_{md}$ | $\times$ 1.388 | $\times$ 1.444 | $76.74 \pm 0.05$ | $1.372 \pm 0.001$ | $0.016 \pm 0.002$ | $0.445 \pm 0.000$ | $3.437 \pm 0.036$ |
| + 3 Bridge$_{md}$ | $\times$ 1.582 | $\times$ 1.666 | $76.85 \pm 0.05$ | $1.353 \pm 0.001$ | $\mathbf{0.015} \pm 0.001$ | $0.443 \pm 0.000$ | $3.967 \pm 0.043$ |
| + 4 Bridge$_{md}$ | $\times$ 1.776 | $\times$ 1.888 | $\mathbf{76.96} \pm 0.03$ | $\mathbf{1.342} \pm 0.001$ | $0.018 \pm 0.001$ | $\mathbf{0.441} \pm 0.000$ | $\mathbf{4.450} \pm 0.062$ |
| DE-2 | $\times$ 2.000 | $\times$ 2.000 | $77.20 \pm 0.07$ | $1.456 \pm 0.004$ | $0.022 \pm 0.002$ | $0.450 \pm 0.002$ | 2.000 |

**Table 12:** Full result of performance improvement of the ensemble by adding type II bridges on ImageNet dataset. We use same settings as described in Table 4.

| Model | FLOPs ($\downarrow$) | #Params ($\downarrow$) | ACC ($\uparrow$) | NLL ($\downarrow$) | ECE ($\downarrow$) | BS ($\downarrow$) | DEE ($\uparrow$) |
|---|---|---|---|---|---|---|---|
| DE-2 | $\times$ 2.000 | $\times$ 2.000 | $77.20 \pm 0.07$ | $0.880 \pm 0.002$ | $0.013 \pm 0.001$ | $0.317 \pm 0.001$ | 2.000 |
| + 1 Bridge$_{md}$ | $\times$ 2.243 | $\times$ 2.256 | $\mathbf{77.43} \pm 0.05$ | $\mathbf{0.870} \pm 0.001$ | $\mathbf{0.012} \pm 0.000$ | $\mathbf{0.314} \pm 0.000$ | $\mathbf{2.564} \pm 0.046$ |
| + 1 Bezier | $\times$ 3.000 | $\times$ 3.000 | $77.65 \pm 0.08$ | $0.861 \pm 0.001$ | $0.011 \pm 0.001$ | $0.311 \pm 0.001$ | $3.059 \pm 0.082$ |
| DE-3 | $\times$ 3.000 | $\times$ 3.000 | $77.64 \pm 0.04$ | $0.862 \pm 0.001$ | $0.013 \pm 0.001$ | $0.311 \pm 0.000$ | 3.000 |
| + 1 Bridge$_{md}$ | $\times$ 3.243 | $\times$ 3.256 | $77.76 \pm 0.07$ | $0.856 \pm 0.001$ | $\mathbf{0.012} \pm 0.001$ | $0.310 \pm 0.000$ | $3.559 \pm 0.038$ |
| + 2 Bridge$_{md}$ | $\times$ 3.486 | $\times$ 3.512 | $77.82 \pm 0.07$ | $0.853 \pm 0.000$ | $\mathbf{0.012} \pm 0.001$ | $0.309 \pm 0.000$ | $3.850 \pm 0.069$ |
| + 3 Bridge$_{md}$ | $\times$ 3.729 | $\times$ 3.768 | $\mathbf{77.92} \pm 0.06$ | $\mathbf{0.851} \pm 0.001$ | $\mathbf{0.012} \pm 0.001$ | $\mathbf{0.308} \pm 0.000$ | $\mathbf{4.010} \pm 0.063$ |
| + 3 Bezier | $\times$ 6.000 | $\times$ 6.000 | $78.30 \pm 0.05$ | $0.834 \pm 0.001$ | $0.012 \pm 0.000$ | $0.303 \pm 0.001$ | $7.821 \pm 0.391$ |
| DE-4 | $\times$ 4.000 | $\times$ 4.000 | $77.87 \pm 0.04$ | $0.851 \pm 0.001$ | $0.012 \pm 0.001$ | $0.308 \pm 0.000$ | 4.000 |

**Table 13:** FLOPs, $R^2$ scores, and model performance metrics between type I bridge network and CNN$^{(ft)}$ $\times n$ + CNN models with various sizes on CIFAR-100. Here, $\times n$ denotes the number of convolution layers to use as a frontal feature extractor CNN$^{(ft)}$, and CNN denotes the same architecture used in the type I bridge network. $R^2$ scores are measured with respect to the target Bezier $r = 0.5$.

| Model | FLOPs ($\downarrow$) | $R^2$ ($\uparrow$) | ACC ($\uparrow$) | NLL ($\downarrow$) | ECE ($\downarrow$) | BS ($\downarrow$) |
|---|---|---|---|---|---|---|
| Type I Bridge | $\times$ 0.211 | **0.786** $\pm$0.003 | **72.18** $\pm$0.19 | **1.016** $\pm$0.004 | 0.031 $\pm$0.002 | **0.379** $\pm$0.001 |
| CNN$^{(ft)}$ $\times 2$ + CNN | $\times$ 0.230 | 0.626 $\pm$0.007 | 63.45 $\pm$0.71 | 1.318 $\pm$0.019 | 0.015 $\pm$0.002 | 0.481 $\pm$0.007 |
| CNN$^{(ft)}$ $\times 4$ + CNN | $\times$ 0.373 | 0.682 $\pm$0.005 | 67.39 $\pm$0.42 | 1.166 $\pm$0.015 | 0.013 $\pm$0.001 | 0.436 $\pm$0.004 |
| CNN$^{(ft)}$ $\times 6$ + CNN | $\times$ 0.515 | 0.685 $\pm$0.006 | 67.43 $\pm$0.55 | 1.156 $\pm$0.019 | 0.014 $\pm$0.001 | 0.433 $\pm$0.006 |
| CNN$^{(ft)}$ $\times 8$ + CNN | $\times$ 0.648 | 0.701 $\pm$0.001 | 68.36 $\pm$0.35 | 1.121 $\pm$0.009 | **0.012** $\pm$0.002 | 0.422 $\pm$0.004 |

### B.3 COMPARISON WITH THE TYPICAL KNOWLEDGE DISTILLATION

We note that mimicking the original function defined by deep neural networks using relatively cheaper networks reminds of Knowledge Distillation (KD) (Hinton et al., 2015), and thus one can think of the proposed approach as a special instance of the knowledge distillation. However, the proposed bridge network differs fundamentally from KD in that; 1) it uses a very small network that cannot be properly trained with a typical distillation procedure, and 2) while KD builds a student mapping input to the output, ours reuses outputs from the models related to the target function to be mimicked, and this actually plays a key role in the function matching.

Here, we empirically validate the claim. Specifically, Table 13 compares bridge networks mimicking output probabilities from $\boldsymbol{\theta}_{1,2}^{(be)}$ 1) when it takes inputs $\boldsymbol{x}$ as in the typical knowledge distillation framework, and 2) when it takes outputs from $\boldsymbol{\theta}_1$ and $\boldsymbol{\theta}_2$ as we proposed. The former consistently underperforms compared to the latter, even if we introduce some frontal convolutional layers for dealing with image inputs. It indicates that the typical knowledge distillation procedure suffers from an insufficient capacity of the bridge network, while our proposed method does not. Consequently, our proposed method, which reuses informative outputs from $\boldsymbol{\theta}_1$ and $\boldsymbol{\theta}_2$, is distinct from the typical knowledge distillation when the capacity of the bridge network is limited.

