# OpenReview forum: "Traversing Between Modes in Function Space for Fast Ensembling"
_ICLR.cc/2023/Conference — Submitted to ICLR 2023_

### Official Review · Reviewer_Kw57 · 2022-10-20

**Confidence:** 4
**Correctness:** 3
**Technical Novelty And Significance:** 3
**Empirical Novelty And Significance:** 3
**Recommendation:** 5

**Clarity, Quality, Novelty And Reproducibility:**

Clarity: High; the paper is overall well-written and easy to understand.

Quality: See strengths and weaknesses above.

Novelty: OK; the main idea is novel and interesting, but it is a rather complex approach for which it is not clear to me whether it would be widely applicable in practice among the large number of approaches that aim to improve the accuracy/efficiency ratio.

Reproducibility: High; the authors provide the training code and configurations used.

**Strength And Weaknesses:**

**Strengths**
* The paper proposes a novel idea for approximating gains of ensembles at lower computational cost (at inference time).
* The presented results are interesting and show that the presented method works (for small data set and classifier sizes).
* The paper is generally easy to follow.

**Weaknesses**
* The method is relatively complex in relation to the performance gains that can be achieved.
* A direct comparison to other methods in terms of the accuracy/computation frontier is missing. It seems that the numbers are reasonable, but e.g. the range of 63% to 67% on Tiny-INet (Table 4) is in the area that can be also reached by a single ResNet-18 according to [Papers with Code](https://paperswithcode.com/sota/image-classification-on-tiny-imagenet-1). Also, there are no direct comparisons to other "efficient ensemble" approaches.
* FLOPS alone are often not a sufficient measure of model efficiency, see e.g. [The Efficiency Misnomer](https://arxiv.org/pdf/2110.12894.pdf)

Further comments:

It was not clear to me after reading the paper at which point in the base networks the z_{i} are extracted. I think this should be clarified (or maybe I missed it?)

I did not see an explanation for the term "Bridge-S" in the paper. (Maybe I missed it?) I guess it refers to a "small" Bridge model, but I did not see what exact configuration "Bridge" and "Bridge-S" refer to.

I found the discussion of knowledge distillation a bit distracting from the main idea. Especially the experimental setting described in the last paragraph of Sec 5.2 has a form that is not super-convincing in my opinion.

It seems to me that the most interesting setting among those discussed is the DE-1 + n bridges. In that case it would actually be possible to try to approximate the r=1 responses of the ensemble (i.e. approximate the outputs of the other ensemble members from the intermediate feature). If this worked similarly well, the whole construction of the Bezier interpolation could be avoided. I think this might be an interesting ablation experiment, but maybe not feasible to perform during the rebuttal.

Overall, I found Sec.4 rather short, given that this area of research is currently quite active. For example, the following papers could be considered relevant here. (I'm not requesting that all of these should be cited, just to give a few examples).
* [Batch Ensembles](https://arxiv.org/pdf/2002.06715.pdf)
* [Ensembles with Shared Representations](https://arxiv.org/pdf/2103.03934.pdf)
* [Evaluating Scalable BDL](https://arxiv.org/pdf/1906.01620.pdf)

Minor points or typos not affecting the evaluation:
* It seems that Eq. (3) and (1) can be combined into a shorter version of the two equations.
* p.3 line ~5: parameters ... achieves -> achieve
* p.3 line ~5: add them to [the/an] ensemble
* p.3 line ~8: "these strategy provide" -> this strategy provides
* p.4 Eq. (9): last "i" should be a "j"
* Algorithm 1 looks like a fairly standard training loop, unless I overlooked something. Maybe removing it would enable moving parts of the Appendix into the main paper.
* p.4 bottom: "when the bridge network does not step toward the Bezier predictions" - I did not fully understand what this means.
* Table 2: The FLOPS are given as a multiple. I assume this is as a multiple of a base network?
* p.7 ~center: "achieve decent R^2 score" -> "a score" or "scores"

**Summary Of The Paper:**

The paper proposes a new way of using an ensemble of neural networks to construct a new classifier that achieves better accuracy at a lower number of FLOPS (at inference time). This is achieved by creating so-called "bridge-classifiers" that approximate the output of an interpolating classifier between pairs of ensemble-base classifiers. The interpolating classifiers are picked from the low-loss Bezier curve in parameter space between the pair of ensemble members. The approximation classifiers are faster than computing more ensemble members because they are smaller classifiers trained on intermediate features that have already been computed in the ensemble. The paper provides experiments on (small) image datasets (CIFAR and Tiny-ImageNet) and (small) classifiers of the ResNet family.

**Summary Of The Review:**

The paper contains interesting approaches and results. However, there are also some weaknesses and limitations. I therefore think that the paper in its current form is marginally below the acceptance threshold, but I would be happy to increase my score during the discussion period if some of the concerns can be addressed by the authors.

---

> ### Author Response · Authors · 2022-11-11
> **Reponse for Reviewer Kw57 #1**
>
> Thank you for your detailed review and interest in our paper. We would like to answer the issues and the minor points.
>
>  > The method is relatively complex in relation to the performance gains that can be achieved.
>
> We respectfully disagree with this; we believe the performance gain achieved by our method is worth the additional training procedure. Moreover, we think that training bridge networks do not require difficult techniques or severe overheads.
>
> > A direct comparison to other methods in terms of the accuracy/computation frontier is missing. It seems that the numbers are reasonable, but e.g. the range of 63% to 67% on Tiny-INet (Table 4) is in the area that can be also reached by a single ResNet-18 according to Papers with Code. Also, there are no direct comparisons to other "efficient ensemble" approaches.
>
> As we replied to Reviewer bBzc, existing methods (Huang et al., 2017; Garipov et al., 2018; Benton et al., 2021; Wortsman et al., 2021) have worse performance compared to deep ensembles while the inference cost remains the same as deep ensembles. Again, we believe comparing ours with the deep ensemble is sufficient to demonstrate the efficacy of our proposed approach.
>
> Also, there may be some performance gaps between ours and the reported results since we used Filter Response Normalization (FRN; Singh and Krishnan, 2020) instead of Batch Normalization (BN; Ioffe and Szegedy, 2015). As we specified in Appendix A.1, FRN has the double advantage of both (1) avoiding recomputation of BN statistics along the subspaces (Garipov et al., 2018) and (2) giving us a clear interpretation of the parameter space (Wenzel et al., 2020; Izmailov et al., 2021). How we define such normalization operation is just architectural details, and thus we see no reason why our method would not work with the conventional BN layers.
>
> > FLOPs alone are often not a sufficient measure of model efficiency, see e.g. The Efficiency Misnomer.
>
> We agree that even if the number of floating-point operations (FLOPs) is a representative measure for quantifying the inference cost, it cannot fully explain the model's efficiency. As you mentioned, Dehghani et al. (2020) pointed out that there can be contradictions between commonly used metrics (e.g., FLOPs, the number of parameters, and throughput) and suggested refraining from reporting results using just a single one. Following this suggestion, the revised version of our paper now presents both FLOPs and the number of parameters. Thank you for your constructive comments.
>
> > It was not clear to me after reading the paper at which point in the base networks the z_{i} are extracted. I think this should be clarified (or maybe I missed it?)
>
> The “$z_{i}$” denotes a set of features from one or more blocks in the base network. We practically choose features of the second or the third to last block. And we use features from one block for relatively simple tasks (e.g. CIFAR-10/100) and use features from two blocks for relatively hard tasks (e.g. Tiny ImageNet and ImageNet). We added these details in the appendix.
>
> > I did not see an explanation for the term "Bridge-S" in the paper. (Maybe I missed it?) I guess it refers to a "small" Bridge model, but I did not see what exact configuration "Bridge" and "Bridge-S" refer to.
>
> Thanks for pointing out this. We added the description of each model and changed the notation in the table; “Bridge-S” to “Bridge_{sm}” for the small version whose FLOPs is less than 15% of a single base model, and “Bridge” to “Bridge_{md}” for the medium version whose FLOPs is more than 15%.
>
> > I found the discussion of knowledge distillation a bit distracting from the main idea. Especially the experimental setting described in the last paragraph of Sec 5.2 has a form that is not super-convincing in my opinion.
>
> Thank you for your constructive comment that improves the presentation of our work. We fully agree with that opinion (i.e., the knowledge distillation part may distract from the main idea), and we decide to move the knowledge distillation part to the appendix. We believe that this revision improves the presentation of our work and further highlights the main idea.

---

> > ### Author Response · Authors · 2022-11-11
> > **Response for Reviewer Kw57 #2**
> >
> > > It seems to me that the most interesting setting among those discussed is the DE-1 + n bridges. In that case it would actually be possible to try to approximate the r=1 responses of the ensemble (i.e. approximate the outputs of the other ensemble members from the intermediate feature). If this worked similarly well, the whole construction of the Bezier interpolation could be avoided. I think this might be an interesting ablation experiment, but maybe not feasible to perform during the rebuttal.
> >
> > The fundamental design principle of our proposed bridge models is that there exists a correspondence between two points on the low-loss Bezier curve (not only in the weight space but also in the function space). Indeed, the experiments in Section 5.2 verify that the bridge models can predict the function output of the target model (i.e., r=0.5) from the base model (i.e., r=0.0; or r=0.0 and r=1.0 for the type II bridge). The key ingredient here is that the target model is closely related to the base model in terms of mode connectivity in the weight space.
> >
> > In this regard, directly approximating the r=1.0 response from the r=0.0 response without such connectivity is much more challenging due to the weak relevance between the two models. Concretely, Table A3 demonstrates whether it is possible to accomplish this in the same recipe we proposed. The R² scores show the bridge models have some trouble approximating the r=1.0 response, leading to the performance degradation of the final ensemble results. To investigate how we predict the r=1.0 response from the r=0.0 would be interesting future work.
> >
> > Table A3. Comparison of the performance improvement and the correspondence between type I bridge networks which approximate the model from the model located at the midpoint of the Bezier curve (r=0.5) and type I bridge networks which approximates the base model located at another endpoint (r=1.0). The results are evaluated once on the CIFAR-100 dataset. R² scores of each bridge are measured on corresponding approximation targets and then averaged.
> >
> > Model               | FLOPs (↓) | #Params (↓) | ACC (↑) | NLL (↓) | ECE (↓) | BS (↓) | R² (↑)
> > --------------------|-----------|-------------|---------|---------|---------|--------|-------
> > ResNet (DE-1)       |   x 1.000 |     x 1.000 |   73.13 |   1.104 |   0.046 |  0.380 |      -
> > |||||||
> > \+ 1 Bridge (r=0.5) |   x 1.211 |     x 1.161 |   74.29 |   0.972 |   0.026 |  0.357 |  0.788
> > \+ 2 Bridge (r=0.5) |   x 1.422 |     x 1.322 |   74.43 |   0.943 |   0.026 |  0.352 |  0.788
> > \+ 3 Bridge (r=0.5) |   x 1.633 |     x 1.483 |   74.71 |   0.933 |   0.025 |  0.351 |  0.788
> > \+ 4 Bridge (r=0.5) |   x 1.844 |     x 1.644 |   74.86 |   0.926 |   0.024 |  0.350 |  0.788
> > |||||||
> > \+ 1 Bridge (r=1.0) |   x 1.211 |     x 1.161 |   73.87 |   1.012 |   0.028 |  0.365 |  0.734
> > \+ 2 Bridge (r=1.0) |   x 1.422 |     x 1.322 |   73.97 |   0.992 |   0.028 |  0.363 |  0.738
> > \+ 3 Bridge (r=1.0) |   x 1.633 |     x 1.483 |   74.15 |   0.979 |   0.028 |  0.361 |  0.738
> > \+ 4 Bridge (r=1.0) |   x 1.844 |     x 1.644 |   74.03 |   0.970 |   0.030 |  0.360 |  0.739
> >
> > > Overall, I found Sec.4 rather short, given that this area of research is currently quite active. For example, the following papers could be considered relevant here. (I'm not requesting that all of these should be cited, just to give a few examples). E.g., Batch Ensembles, Ensembles with Shared Representations, and Evaluating Scalable BDL.
> >
> > Following your suggestion, we revised Section 4 to offer additional literature showing advances in this area of research. Thank you for the comment that makes our paper solid.

---

> > > ### Comment · Reviewer_Kw57 · 2022-11-17
> > > **Thank you for the response**
> > >
> > > I would like to thank the authors for the responses to the reviews and for the changes made in the paper. Some of my questions have been clarified and some concerns addressed.

---

> > > > ### Author Response · Authors · 2022-11-18
> > > > **Thanks for your positive response**
> > > >
> > > > We would like to ask if your concerns have been resolved enough to consider raising your score. Or, if there's something still missing from our answer, we would really appreciate it if you could let us know.

---

### Official Review · Reviewer_tqBD · 2022-10-24

**Confidence:** 4
**Correctness:** 3
**Technical Novelty And Significance:** 3
**Empirical Novelty And Significance:** 2
**Recommendation:** 5

**Clarity, Quality, Novelty And Reproducibility:**

The idea of the proposed deep ensembling is novel and the clarity and originality of the work are good. However, the experiments are conducted on small-scale datasets, which is not convincing to the community.

**Strength And Weaknesses:**

Strengths:
(1)	The idea of building bridges between different models for ensembling is interesting. It re-uses the features and only focuses on classification heads. Therefore, it would reduce the inference cost compared with directly ensembling different models.
(2)	The experiments on several datasets show the effectiveness of the proposed framework.


Weaknesses:
(1)	I concern only considering classification would not lead to diverse solutions to ensembling. As described in the paper, the bridges are light-weight MLPs.
(2)	The paper pointed out “these methods (the existing methods) do not scale well for complex large-scale datasets or require network capacity”. However, this paper also DO NOT scale up on complex large-scale datasets, such as ImageNet-1k. Since the experimental platform is on 8 TPUs, it is expected to conduct on large-scale datasets such as ImageNet-1K, large-scale backbones (or SOTA backbones) such as transformers. The experiments on CIFAR-10, CIFAR-100 or tiny ImageNet are not convincing to me.
(3)	Minor comments, Page 1: the modes, three models, two modes? Mode or models?



**Summary Of The Paper:**

This paper proposes a new ensembling framework to reduce inference cost and boost the performance. The proposed method build additional light-weight head as bridges to ensemble different runs. The experiments show the effectiveness of the proposed method on CIFAR-10, CIFAR-100, and tiny-ImageNet.

**Summary Of The Review:**

(1)The idea of the proposed deep ensembling is novel and interesting.
(2) The experiments on CIFAR-10, CIFAR-100, and tiny-ImageNet shows the improvement, but do not scale up to large-scale datasets and large-scale backbones (or SOTA backbones) .

---

> ### Author Response · Authors · 2022-11-11
> **Response for Reviewer tqBD**
>
> Thank you for your review. We would like to answer your comments.
>
> > I concern only considering classification would not lead to diverse solutions to ensembling. As described in the paper, the bridges are light-weight MLPs.
>
> We apologize for the confusing typo in the conclusion section (MLP → CNN). We actually use CNN to construct the bridge network. For the first question (concern only considering classification…), we didn’t actually understand your question, so it would be appreciated if you further clarify it.
>
> > The paper pointed out “these methods (the existing methods) do not scale well for complex large-scale datasets or require network capacity”. However, this paper also DO NOT scale up on complex large-scale datasets, such as ImageNet-1k. Since the experimental platform is on 8 TPUs, it is expected to conduct on large-scale datasets such as ImageNet-1K, large-scale backbones (or SOTA backbones) such as transformers. The experiments on CIFAR-10, CIFAR-100 or tiny ImageNet are not convincing to me.
>
> We also agree with the comments that such large-scale experiments make our paper convincing, and we present additional experimental results on ImageNet in the general response. We believe the ImageNet results demonstrate the scalability of our approach. Regarding the transformers, it has been reported that the transformer models are not easily connected by low-loss subspaces as for the typical feed-forward neural networks (Benton et al., 2021), so we believe it would be hard to test our approach during the rebuttal period. However, we definitely agree that expanding the scope of our method to transformer-related models would be an important future research direction.
>
> > Minor comments, Page 1: the modes, three models, two modes? Mode or models?
>
> The “mode” in the mode connectivity context denotes the local optima on the loss surface of a deep neural network. And the “model” denotes the architecture combined with a certain parameter that can execute the forward pass to compute the output. Given this, the sentence “an ensemble of three models (two modes and one in the connecting subspace)” on Page 1 means that an ensemble is constructed from three parameters collected from three different locations; one parameter from a subspace connecting two modes, and two parameters from the modes on each side. We added what the “mode” is and clarified the sentence.

---

> > ### Comment · Reviewer_tqBD · 2022-11-20
> > **In-depth analysis about computational cost**
> >
> > I would like to thank the authors' responses to my comment.
> >
> > According to the general response. ResNet-50 (DE-1) achieves about 75.85 in accuracy and requires x1.00 flops and  x1.00 params. The + 1 Bridge achieves 76.57 and requires x 1.194 flops and x 1.222 params. This indicates the proposed method needs $\sim$20\% computational cost to achieve 0.72\% acc gain.
> >
> > ------------
> >
> > ResNet (DE-1)	x 1.000	x 1.000	75.85 ± 0.06	0.936 ± 0.003	0.019 ± 0.001	0.333 ± 0.001	1.000	1.000
> >
> > \+ 1 Bridge	x 1.194	x 1.222	76.57 ± 0.02	0.906 ± 0.001	0.012 ± 0.001	0.324 ± 0.000	1.527 ± 0.054	1.535 ± 0.042
> >
> > ------------
> >
> > The main improvement could be from the increase of CNN layers (params/flops). One way to fair comparison is to increase the baseline ResNet (DE-1) from ResNet-50 to ResNet-60 or even ResNet-70 to keep similar parameters/flops with \+ 1 Bridge.
> >
> > I observed that the Keras codebase (https://keras.io/api/applications/) yields 76.4\%-74.9\%=1.5\% improvement in accuracy from ResNet50 to ResNet101 while needs (44.7M-25.6M)/25.6M=74% more params. Therefore, I am afraid that simply adding layers to the baseline (ResNet (DE-1)) may also achieve similar results.

---

> > > ### Author Response · Authors · 2022-11-20
> > > **Response to additional comments by Reviewer tqBD**
> > >
> > > Thanks for your constructive response to our general comment.
> > >
> > > > The main improvement could be from the increase of CNN layers (params/flops). One way to fair comparison is to increase the baseline ResNet (DE-1) from ResNet-50 to ResNet-60 or even ResNet-70 to keep similar parameters/flops with + 1 Bridge. I observed that the Keras codebase yields 76.4%-74.9%=1.5% improvement in accuracy from ResNet50 to ResNet101 while needs (44.7M-25.6M)/25.6M=74% more params. Therefore, I am afraid that simply adding layers to the baseline (ResNet (DE-1)) may also achieve similar results.
> > >
> > > As we mentioned in [the thread in the general comment to Reviewer bBzc (item 1. in the reply)](https://openreview.net/forum?id=cS45VNtZLW&noteId=uXuJpKiomX), an ensemble of small models performs about the same or better performance than a large single model (Wang et al., 2022). Specifically, an ensemble of two ResNet-50 models performs almost the same as a single ResNet-101 model with almost the same FLOPs (Figure 2(b) of the paper; DE-2 of ResNet-50 shows ACC 77.8 and FLOPs 7.0B, single ResNet-101 shows ACC 77.9 and FLOPs 7.2B). Both the Keras example you mentioned and the paper (Wang et al., 2022) have slightly different model structures from ours, so it is difficult to make a one-to-one comparison with the specific numbers from other sources; we use FRN instead of BN as we explained in the appendix. However, we also expect the DEs to show at least the performance of a single large model.
> > >
> > > For the above reasons, we believe that only comparison with DE can sufficiently show that our model is efficient. However, as Reviewer bBzc also noted, we agree that the comparison with other similar methods will strengthen our argument. We would like to add this comparison in a later revision.
> > >
> > > **References**
> > > - X. Wang, D. Kondratyuk, E. Christiansen, K. M. Kitani, Y. Movshovitz-Attias, and E. Eban. Wisdom of Committees: An Overlooked Approach To Faster and More Accurate Models. In ICLR, 2022.

---

### Official Review · Reviewer_bBzc · 2022-10-25

**Confidence:** 3
**Correctness:** 3
**Technical Novelty And Significance:** 2
**Empirical Novelty And Significance:** 2
**Recommendation:** 5

**Clarity, Quality, Novelty And Reproducibility:**

Proper illustrations has been given for the effects of the method. It would be appreciated if heatmap (loss landscape) can be given to facilitated comparisons in this line of work.

**Strength And Weaknesses:**

Strength
The work is well motivated and the two proposed bridge networks method are well crafted.

Weakness
1. there is a key alternative not compared, in Learning Neural Network Subspaces a method is proposed to train models that can be linearly interpolated. Though it is not easy to compare the approaches, one is altering the training process and the other is more about adding an ancillary equipment, the two approaches can be evaluated in the same coordinate system with inference cost and accuracy on two axes.
1. the method has not been verified on large scale datasets, though it criticized literature with "these methods do not scale well for complex large-scale datasets"

**Summary Of The Paper:**

The paper proposes two methods to reduce the inference cost of Deep Ensemble (a collection of ensemble models of the same model structure). Bézier Curve is fitted on the extracted features by a neural network (bridge neural network), so that "interpolated" inference results can be cheaply approximated by the bridge NN.

**Summary Of The Review:**

The paper is well motivated and easy to follow, but still lacks solid grounding for its claims.

---

> ### Author Response · Authors · 2022-11-11
> **Response for Reviewer bBzc**
>
> Thank you for your review. We would like to answer the issues kindly raised by the reviewer.
>
> > There is a key alternative not compared, in Learning Neural Network Subspaces a method is proposed to train models that can be linearly interpolated. Though it is not easy to compare the approaches, one is altering the training process and the other is more about adding an ancillary equipment, the two approaches can be evaluated in the same coordinate system with inference cost and accuracy on two axes.
>
> Based on previous literature arguing the existence of the low-loss subspaces in the weight space for deep neural networks (Garipov et al., 2018; Draxler et al., 2018), Wortsman et al. (2021) further proposed an algorithm for finding such subspaces in a single training run. However, it still requires multiple forward passes for ensembling, and the inference cost remains as same as deep ensembles. Apart from Wortsman et al. (2021), several works also proposed training-efficient ensembling methods (Huang et al., 2017; Garipov et al., 2018; Benton et al., 2021), but their performances are worse than deep ensembles when the inference cost gets in line. In other words, such baselines will be located in the worse region of the same coordinate system (e.g., the bottom right position in the first plot of Figure 4) and do not provide meaningful comparisons beyond deep ensembles. Thus, we believe comparing ours with the deep ensemble is sufficient to demonstrate the efficacy of our proposed approach.
>
> > The method has not been verified on large scale datasets, though it criticized literature with "these methods do not scale well for complex large-scale datasets"
>
> Please refer to the general response containing ImageNet results. We believe that now our proposed method has been verified on large-scale datasets. Thank you for your suggestion that makes our paper solid.
>
> > Proper illustrations has been given for the effects of the method. It would be appreciated if heatmap (loss landscape) can be given to facilitated comparisons in this line of work.
>
> Note that the proposed bridge models are designed to approximate the midpoint of the Bezier curve in the function space, not the weight space. We cannot visualize the bridge models in such loss landscape plots along with the base networks (e.g., Figure 1 in Garipov et al. (2018)) since the bridge models have different architecture. Instead, experimental results in Section 5.2 demonstrate that the bridge models effectively approximate the function outputs of the midpoint of the Bezier curve.

---

### Official Review · Reviewer_iSwW · 2022-10-25

**Confidence:** 4
**Clarity, Quality, Novelty And Reproducibility:** see above
**Correctness:** 3
**Technical Novelty And Significance:** 3
**Empirical Novelty And Significance:** 2
**Recommendation:** 5

**Strength And Weaknesses:**

### Strength

* Reducing the inference cost of DE is an important topic. This paper is well-motivated.
* The presentation of this paper is clear and easy to follow.
* Extensive experiments demonstrate the effectiveness of the proposed two types of bridge networks on TinyImageNet and Cifar.

### Weaknesses
* In Table 4, why do more bridge models lead to better results?
* What's the functional difference between type I and II bridge models according to theoretical and empirical results? Is there any conclusion we can reach about how to choose types I and II?
* Are there any important/sensitive hyper-parameters during the training of bridge models? How to determine the optimal number of bridge models in practice?
* I am curious about how the proposed methods perform on large-scale datasets like ImageNet.

**Summary Of The Paper:**

This paper aims to address an important drawback of Deep Ensemble, the inference cost of executing multiple models. The intuition behind the proposed method is that the outputs in the function subspace can be estimated from the modes without having to forward the actual parameters on the subspace. Based on this intuition, an additional lightweight network is trained as a bridge network to predict the outputs from the connecting subspace.

**Summary Of The Review:**

I have some concerns about
1) fundamental difference between type I and II bridge models
2) Why number of bridge models impact
3) generalization of this method to the large-scale datasets

I am willing to raise my score if the above concerns are addressed.

---

> ### Author Response · Authors · 2022-11-11
> **Response for Reviewer iSwW**
>
> Thank you for your review. We would like to answer your questions.
>
>
> > What’s the functional difference between type I and II bridge models according to theoretical and empirical results? I have some concerns about fundamental difference between type I and II bridge models.
>
> Both types I and II bridges approximate the model located at the midpoint of the Bezier curve with a small inference cost. While the type II bridge requires both outputs of models located at two endpoints of the Bezier curve, the type I bridge only requires one. Since we provide more information about the Bezier subspace to the type II bridge, it will approximate the midpoint of the Bezier curve more accurately than the type I bridge. Indeed, the experimental results presented in Section 5.2 show that the type II bridge produces more similar output to the target model (i.e., the midpoint of the Bezier curve) than the type I bridge. Nevertheless, it is worth investigating the type I bridge since its strength lies in the possibility of scaling up a single model. To be specific, Section 5.3.1 shows that type I bridges can progressively enhance the single model to be stronger than DE-{1,2,3} with a relatively lower cost.
>
> > In Table 4, why do more bridge models lead to better results? I have some concerns about why number of bridge models impact.
>
> Here, each bridge approximates the models on different Bezier curves between a single mode and others (i.e., Bezier curves between modes A-B, A-C, and so on where A, B, and C are different modes.), not the models on a single Bezier curve. Adding more bridge networks introduces more diverse outputs to the ensembles. We clarified this point in the revised version. Thank you for your constructive comments.
>
> > Are there any important/sensitive hyper-parameters during the training of bridge models?
>
> We introduced two additional hyperparameters for training bridge models in the main text; 1) the regularization scale λ and 2) the mixup coefficient α. Since the training error of the base network is near zero for the family of residual networks on CIFAR-10/100, given a training input without any modification, the base network and the target network (the one on the Bezier curve) will produce almost identical outputs, so the bridge trained with them will just copy the outputs of the base network. To prevent this, we perturb the inputs via mixup, and regularize the bridge to produce outputs different from the ones computed from the base models. On the other hand, for the datasets such as ImageNet where the models fail to achieve near zero training errors, the base network and the target networks are already distinct enough, so we found that the bridge can be trained easily without such tricks (i.e., we used λ=0.0 and α=0.0).  Thus, based on the training error of the base model, we can roughly determine two hyperparameters λ and α in practice (i.e., use λ>0 and α>0 values when the training error of the base network is near zero). For the other hyperparameters, we used the same learning rate, momentum, and weight decay values as when training base networks.
>
> > How to determine the optimal number of bridge models in practice?
>
> Like deep ensembles, our method also saturates when the number of bridges gets larger. However, note that the proposed bridge model has less pressure on increasing the ensemble size. Thus, one practical choice in a resource-limited environment is adding as many bridges as possible until they fully utilize the allowed computation budget.
>
> > I am curious about how the proposed methods perform on large-scale datasets like ImageNet.
>
> We are glad to present large-scale results on ImageNet during the rebuttal period. Please refer to the general response for more details. Our proposed method is still beneficial regardless of the scale of the dataset.

---

> > ### Comment · Reviewer_iSwW · 2022-11-15
> > **Thanks for the response**
> >
> > Thanks for the authors' response. For the second question, I appreciate the answer from the authors which makes this question clear. For the first question, I can get the point of the answer but could still be confused when I need to select type I/II bridge in practice. In terms of the hyper-parameters, I think some training hyper-parameters like learning rate and type of optimizer of bridge model should also be counted. Given the number of hyper-parameters introduced and the performance improvement on ImageNet, the practicality of the proposed could be further improved.

---

> > > ### Author Response · Authors · 2022-11-16
> > > **Response to additional comments by Reviewer iSwW**
> > >
> > > Thanks for the comments.
> > >
> > > > For the first question, I can get the point of the answer but could still be confused when I need to select type I/II bridge in practice.
> > >
> > > Although type II bridges usually perform better than type I (due to the rich information that came from two endpoints of the Beizer curve), they presume the feasibility of multiple base networks during inference. Thus, in practice, it is encouraged to use type II bridges only if we are allowed to forward multiple base networks (if not, we should use type I bridges). If the available FLOPs and memory are between DE-1 and DE-2, only Type I bridges can be used, because they can work with a single base model (one endpoint). If you have more FLOPs and more memory available than the DE-2, you can use type II bridges. Although type II bridges require base models for both endpoints, they can make more accurate predictions and show higher ensemble gains than type I bridges. This point can be confirmed more clearly by looking at the x-axis (relative FLOPs) of Figure 4 and Figure 5 of the paper, which shows the relative performance compared to the relative FLOPs of type I and type II bridges, and the performance improvement accordingly. In summary, type I bridges can be effectively used in a situation where available relative FLOPs are x1 ~ x2, and type II bridges can be effectively used in a situation where available relative FLOPs are > x2.
> > >
> > > > In terms of the hyper-parameters, I think some training hyper-parameters like learning rate and type of optimizer of bridge model should also be counted. Given the number of hyper-parameters introduced and the performance improvement on ImageNet, the practicality of the proposed could be further improved.
> > >
> > > We agree that the type of optimizer and learning rates should be counted as hyperparameters, but our method is not sensitive to them. We used the same standard hyperparameter settings when training the base models, except for the two additional hyperparameters described in the previous response. We used a common learning rate of 0.1, a cosine learning rate scheduler, and an SGD optimizer to train all base, bezier, and bridge models in all datasets. Weight decay values were set differently for each dataset as we presented in the appendix, but the same value was used for all models in one dataset. The only difference is that the base and bezier models were trained for 200 epochs, and the bridge models were trained for 50 epochs because the bridge models learn relatively quickly. As such, the method of setting the two hyperparameters specially involved in training the bridge model was explained in the previous response, and for other hyperparameters, the same values as when learning the base model can be used as they are. Therefore, from a practical point of view, additional tuning is hardly required. We believe that the performance gain (in terms of DEE) for ImageNet is remarkable, considering the simplicity of our model, and this gain is easily reproduced without heavy hyperparameter tuning.

---

### Author Response · Authors · 2022-11-11
**General Response from Authors**

Thanks for the constructive reviews from all reviewers. We mark the revised text in blue on the paper.


Before the rebuttal period, we conducted ImageNet experiments to show that our method also scales well for large-scale datasets. We use ResNet-50 as a base model and CNN 3 blocks as type I and type II bridge networks. Other settings are the same as we described in the experiment section (Section 5). We summarize the type I results in Table A1 and the type II results in Table A2. We also add the results to the paper.


Table A1. Performance improvement of the ensemble by adding type I bridges to the single base ResNet on ImageNet dataset. FLOPs, #Params, DEE, and Rel. ACC metrics are measured with respect to the base ResNet. Adding one bridge network significantly improves the accuracy metric (≈ x1.535) and the uncertainty metrics (≈ x1.527), and the performance continues to improve as more bridges are added.

​​Model         | FLOPs (↓) | #Params (↓) | ACC (↑)      | NLL (↓)       | ECE (↓)       | BS (↓)        | DEE (↑)       | Rel. ACC (↑)
--------------|-----------|-------------|--------------|---------------|---------------|---------------|---------------|--------------
ResNet (DE-1) |   x 1.000 |     x 1.000 | 75.85 ± 0.06 | 0.936 ± 0.003 | 0.019 ± 0.001 | 0.333 ± 0.001 | 1.000         | 1.000
\+ 1 Bridge   |   x 1.194 |     x 1.222 | 76.57 ± 0.02 | 0.906 ± 0.001 | 0.012 ± 0.001 | 0.324 ± 0.000 | 1.527 ± 0.054 | 1.535 ± 0.042
\+ 2 Bridge   |   x 1.388 |     x 1.444 | 76.74 ± 0.05 | 0.900 ± 0.001 | 0.012 ± 0.001 | 0.322 ± 0.000 | 1.644 ± 0.050 | 1.664 ± 0.070
\+ 3 Bridge   |   x 1.582 |     x 1.666 | 76.85 ± 0.05 | 0.897 ± 0.000 | 0.012 ± 0.000 | 0.321 ± 0.000 | 1.689 ± 0.048 | 1.742 ± 0.069
\+ 4 Bridge   |   x 1.776 |     x 1.888 | 76.96 ± 0.03 | 0.896 ± 0.000 | 0.011 ± 0.001 | 0.321 ± 0.000 | 1.710 ± 0.041 | 1.804 ± 0.055
DE 2          |   x 2.000 |     x 2.000 | 77.20 ± 0.07 | 0.880 ± 0.002 | 0.013 ± 0.001 | 0.317 ± 0.001 | 2.000         | 2.000


Table A2. Performance improvement of the ensemble by adding type II bridges as members to existing DE ensembles on ImageNet dataset. FLOPs, #Params, DEE, and Rel. ACC metrics are measured with respect to corresponding DEs. Type II bridges consistently improve the accuracy and uncertainty metrics of the ensemble before saturation. For all DEs, only adding a few type II bridges dramatically improves the performance.

Model       | FLOPs (↓) | #Params (↓) | ACC (↑)      | NLL (↓)       | ECE (↓)       | BS (↓)        | DEE (↑)        | Rel. ACC (↑)
------------|-----------|-------------|--------------|---------------|---------------|---------------|----------------|--------------
DE 2        |   x 2.000 |     x 2.000 | 77.20 ± 0.07 | 0.880 ± 0.002 | 0.013 ± 0.001 | 0.317 ± 0.001 | 2.000          | 2.000
\+ 1 Bridge |   x 2.243 |     x 2.256 | 77.43 ± 0.05 | 0.870 ± 0.001 | 0.012 ± 0.000 | 0.314 ± 0.000 | 2.564 ± 0.046  | 2.519 ± 0.059
DE 3        |   x 3.000 |     x 3.000 | 77.64 ± 0.04 | 0.862 ± 0.001 | 0.013 ± 0.001 | 0.311 ± 0.000 | 3.000          | 3.000
\+ 1 Bridge |   x 3.243 |     x 3.256 | 77.76 ± 0.07 | 0.856 ± 0.001 | 0.012 ± 0.001 | 0.310 ± 0.000 | 3.559 ± 0.038  | 3.541 ± 0.223
\+ 2 Bridge |   x 3.486 |     x 3.512 | 77.82 ± 0.07 | 0.853 ± 0.000 | 0.012 ± 0.001 | 0.309 ± 0.000 | 3.850 ± 0.069  | 3.808 ± 0.201
\+ 3 Bridge |   x 3.729 |     x 3.768 | 77.92 ± 0.06 | 0.851 ± 0.001 | 0.012 ± 0.001 | 0.308 ± 0.000 | 4.010 ± 0.063  | 4.392 ± 0.355
DE 4        |   x 4.000 |     x 4.000 | 77.87 ± 0.04 | 0.851 ± 0.001 | 0.012 ± 0.001 | 0.308 ± 0.000 | 4.000          | 4.000

---

> ### Comment · Reviewer_bBzc · 2022-11-18
> **Great to see results on ImageNet, but comparisons are perferred**
>
> Since there are considerable amount of increase in flops and #params, can some baseline methods with similar level of increase of flops and #params be included for apple-to-apple comparisons?

---

> > ### Author Response · Authors · 2022-11-18
> > **Response to the comment by Reviewer bBzc**
> >
> > Thanks for your constructive comment.
> >
> > > Since there are considerable amount of increase in flops and #params, can some baseline methods with similar level of increase of flops and #params be included for apple-to-apple comparisons?
> >
> > As we answered earlier, previous methods related to loss landscape were mostly considered in terms of the training cost, still inference cost equal to DE. Other methods related to efficiency in inference time include:
> >
> > 1. Use a single large model with FLOPs similar to the increased FLOPs. However, using an ensemble of small models performs better for similar FLOPs than using a single large model (Wang et al., 2022). Therefore, we expect the performance of DE to be higher than that of a single large model.
> > 2. As a way to efficiently use DE, there is a cascade method (Wang et al., 2022) that dynamically determines the number of ensemble members according to the difficulty of the input. This method efficiently inferences by reducing average FLOPs by reducing the number of members used in situations where an ensemble is not required. However, this method has the disadvantage that the performance of DE acts as an upper bound, requires the same FLOPs as DE in the worst case, and cannot compute ensemble members in parallel. Since this can also be used with our method, applying it will make our method more powerful.
> > 3. Batch Ensemble (Wen et al., 2019) is efficient in terms of the number of parameters, but less efficient than DE in terms of FLOPs because each ensemble member must be computed. Also, the performance is lower than that of DE.
> > 4. Methods using a single model can be used in combination with our method if a Bezier curve exists. Since MIMO (Havasi et al., 2021) is also a method using a single model, we expect further performance improvement by using our method as an extension. This will be interesting future work.
> >
> > However, since DE acts as an upper bound for other methods as above, we think that our method sufficiently proves its performance and efficiency just by comparing it with DE. We agree that adding a direct comparison with other methods in a later revision would make a more clear comparison.
> >
> > **References**
> > - X. Wang, D. Kondratyuk, E. Christiansen, K. M. Kitani, Y. Movshovitz-Attias, and E. Eban. Wisdom of Committees: An Overlooked Approach To Faster and More Accurate Models. In ICLR, 2022.
> > - Y. Wen, D. Tran, and J. Ba. BatchEnsemble: an alternative approach to efficient ensemble and lifelong learning. In ICLR, 2019.
> > - M. Havasi, R. Jenatton, S. Fort, J. Z. Liu, J. Snoek, B. Lakshminarayanan, A. M. Dai, and D. Tran. Training independent subnetworks for robust prediction. In ICLR, 2021.

---

### Decision · Program_Chairs · 2023-01-20

**Decision:**

Reject

**Justification For Why Not Higher Score:**

Reasons mentioned in the meta-review.

**Justification For Why Not Lower Score:**

N/A

**Metareview: Summary, Strengths And Weaknesses:**

This paper looks at the problem of efficient test-time prediction using deep ensembles. Traditional deep ensembles require multiple forward passes (one for each member of the ensemble). This paper presents a method that leverages the idea of mode connectivity and designs a "bridge network" which enables computing the prediction at any intermediate point on the path connecting two modes. Notably, computing the predictions doesn't require doing forward passes on the original networks.

The reviewers appreciated the idea. However, there remained several concerns related to the experiments, such as the complexity of the method not commensurate w.r.t. the performance gains achieved. It also remained unclear how the proposed method is positioned as compared to various other existing methods that have tried to address the accuracy vs efficiency trade-off.

Another aspect that the paper misses is comparison with recent methods that distill a deep ensemble into a single network. The paper talks about the basic knowledge distillation (KD) idea but there is a significant amount of recent work on KD for deep ensembles, such as "Ensemble Distribution Distillation" (Malinin et al, 2022) and follow-up works. The paper would be strengthened by a discussion of (and ideally a comparison with) these methods.

The paper was discussed but in the end, it was felt by all the reviewers that, in the current form, the paper does not meet the acceptance bar. The authors are advised to take into account the feedback from the reviewer, which we hope will strengthen the paper, and resubmit to another venue.